# Extreme precipitation events induce high fluxes of groundwater and associated nutrients to coastal ocean

Marc Diego-Feliu[1], Valentí Rodellas[1], Aaron Alorda-Kleinglass[1], Maarten Saaltink[2,3], Albert Folch[2,3], and Jordi Garcia-Orellana[1,4]

[1]Institut de Ciència i Tecnologia Ambientals, UAB, Bellaterra, E-08193, Spain
[2]Department of Civil and Environmental Engineering, UPC, Barcelona, E-08034, Spain
[3]Hydrogeology Group, UPC-CSIC, Barcelona, E-08034, Spain
[4]Departament de Física, Bellaterra, E-08193, Spain

*Correspondence to*: Marc Diego-Feliu (marc.diego@uab.cat)

**Abstract.** Current Submarine Groundwater Discharge (SGD) studies are commonly conducted under aquifer baseflow conditions, neglecting the influence of episodic events that can significantly increase the supply of nutrients and water. This limits our understanding of the social, biogeochemical, and ecological impacts of SGD. In this study, we evaluated the influence of an extreme precipitation event (EPE) on the magnitude of of SGD. To do so, three seawater sampling campaigns were performed at a Mediterranean ephemeral stream-dominated basin after an extreme precipitation event (~90 mm in few hours) and in baseflow conditions. Results indicate that the groundwater flows after the extreme precipitation event were 1 order of magnitude higher than those in baseflow conditions. SGD induced by extreme precipitation events, which only take place a few days per year, represented up to one third of the annual discharge of groundwater and associated nutrients at the study site. This work accentuates the need to account for episodic increases in the supply of water and nutrients when aiming at providing reliable annual SGD estimates, particularly in the current context of climate change, since the occurrence of such events is expected to increase worldwide.

## 1 Introduction

Submarine groundwater discharge (SGD) - the flow of groundwater from the coastal aquifers to the coastal ocean - is one of the primary processes regulating the transfer of solutes from land to ocean (Santos et al., 2021). The significance of this process at local, regional and global scale stems mainly from its role in modulating the water and chemical budgets of oceans, controlling coastal ecosystems, and contributing to the well-being of coastal societies (Alorda-Kleinglass et al., 2021; Lecher et al., 2015; Luijendijk et al., 2020). In the last three decades, there have been many studies focusing on quantifying SGD and associated solute fluxes in multiple sites across the globe, including coves, bays, estuaries, and entire basins (e.g., Beck et al., 2008; Kwon et al., 2014; Tamborski et al., 2020). However, most of the SGD investigations are conducted under baseflow conditions, that is, in the absence of any meteorological, hydrological, or oceanographical event (e.g., storms, monsoons, sea-

level anomalies) which might significantly impact the magnitude of SGD. Only a few articles have focused on the evaluation of the temporal variations in SGD induced by episodic events (Adyasari et al., 2021; Gonneea et al., 2013; Hu et al., 2006; Rodellas et al., 2020; Sugimoto et al., 2016). Extrapolating from SGD estimates derived under baseflow conditions to obtain

annual fluxes neglects the role of such events, which may represent an important contributor to overall SGD fluxes.

Extreme precipitation events (EPE) represent one of the main natural hazards producing severe societal and economic costs in urban, agricultural, and mountainous areas (Booij, 2002; Camarasa-Belmonte and Soriano-García, 2012; Schumacher, 2017). The meteorological causes of these events include the formation of cyclones, fronts, monsoons, isolated thunderstorms, upslope flow precipitation, and others, and vary from region to region (Kunkel et al., 2012). The social and environmental

consequences also vary geographically and depend on diverse aspects such as topography, soil characteristics, geological setting, land surface use and characteristics, and human-induced changes to the landscape and coastal areas (Schumacher, 2017). Due to the disparate causes and consequences associated with them, there are different ways for defining an 'extreme' precipitation event: some authors define such events as those exceeding an arbitrary threshold of 24h-accumulated rainfall (e.g., Lionello et al., 2006; Meenu et al., 2020); others use the $90^{th}$, $95^{th}$, or $99^{th}$ all-day or wet-day percentile (Pendergrass and

Knutti, 2018). Whilst episodic increases in surface runoff linked to EPE are often well characterized (e.g., Camarasa-Belmonte and Tilford, 2002; Moore, 2007; Rajurkar et al., 2004), little is known about the influence of EPE on SGD-driven water flows and associated solute fluxes to the coastal ocean. Extreme precipitation events may indeed promote aquifer recharge through the infiltration of rainwater (Ramos et al., 2020; Yu et al., 2017), although its effects on piezometric levels (quantitively and temporally) depend on several factors, such as soil composition, geological characteristics, the hydraulic parameters of the

aquifer, and others. Infiltrated water displaces groundwater stored in the aquifer towards the sea, and in some cases may also enhance mixing processes in the coastal aquifer (Anwar et al., 2014; Palacios et al., 2019; Robinson et al., 2018). The significance of EPE on SGD may be exacerbated in areas subject to an arid or semi-arid climate, with scarce and unevenly distributed precipitation, and where extreme events may be the only form of recharge for the aquifer (Taylor et al., 2013). Understanding the role of EPE on SGD-driven water flows and associated solute fluxes is thus fundamental in order to (1)

accurately constrain annual fluxes of any dissolved compound transported by SGD, (2) integrate episodic-induced SGD in global estimates, (3) evaluate the environmental impacts of these episodic events on coastal ecosystems and (4) foresee the role of SGD in the future climate change scenario.

This study evaluates the significance of SGD and the associated nutrient fluxes induced by EPE in a Mediterranean coastal zone, the implications of neglecting EPEs on annual estimates, and its importance in the context of climate change, using Ra

isotopes as tracers of SGD. The study was conducted at a typical Mediterranean ephemeral stream-dominated basin. These areas are characterised by rapid response to precipitation due to their geomorphological features (small catchment areas, sharp slopes, coastal alluvial aquifers, and sporadic surficial torrential courses; Camarasa-Belmonte and Segura Beltrán, 2001) and thus represent an ideal setting for gauging the influence of such events.

## 2 Methods

### 2.1 Study site

Maresme County is a coastal region located to the NE of the city of Barcelona (Spain, western Mediterranean Sea) that extends from the Catalan Littoral Mountain Range to the Mediterranean Sea (Fig. 1). The county has a population density of ca. 1100 hab·km$^{-2}$, and is highly anthropized, with 15% agricultural and 30% urban land use (Rufí-Salís et al., 2019). The geomorphology of the area is structurally associated with the fracturing and sinking of blocks (NW-SE direction), which developed in a set of stream valleys (Catalan Water Agency, 2010). The geology of these valleys is dominated by Quaternary detrital sediments that constitute layers of gravels, sands, and clays, which result from chemical weathering and the dragging of granite materials through torrential courses. These Quaternary deposits constitute different aquifers corresponding with each stream valley (Fig. 1). Simultaneously, each stream valley forms an ephemeral stream. In Maresme County there are around 40 ephemeral streams, which represents a density of 1 ephemeral stream per kilometer of coastline. Annual precipitation in the area ranges from 350 to 930 mm y$^{-1}$ (2015 to 2020) and is mainly governed by EPE occurring in the autumn and spring seasons. Precipitation events with >75 mm d$^{-1}$ (corresponding to the 99$^{th}$ wet-day percentile) are here considered as EPE (Pendergrass and Knutti, 2018) and have a recurrence of ca. 13 months. In this area most of these streams are hydraulically disconnected from their alluvial aquifers and, therefore, surface runoff takes place only after the most significant rain events, which are characterized by its short duration and great intensity. The nature of floods associated to EPEs are well known and have been described in detail especially in grey literature (Cisteró and Camarós, 2014; Riba, 1997). The flood events consist of different stages which take place in few hours (2 to 6) depending on the intensity and duration of the EPE: (1) some minutes after the precipitation has started a thin layer (some cm) of 'dirty' water (from the surroundings) flow towards the sea, (2) after some minutes to hours, a cleaner water mass which carries heavier materials overcomes the first one, (3) the flood level increases progressively towards a maximum discharge rate which remains constant for a short period of time, and (4) the water level decreases gradually until completely disappearing (Cisteró and Camarós, 2014). Water velocities in one of the heaviest precipitation events occurred in the Argentona ephemeral stream (180 mm in 24 h) was calculated to be on the order of 2.7 to 3.8 m s$^{-1}$ (Martín-Vide, 1985). The phreatic level in the lower part of the ephemeral streams is shallow (2 to 3 meters below the ground level) and the materials are highly permeable (Martínez-Pérez et al., 2022). This facilitates the rapid aquifer recharge after an EPE, since infiltrated rainwater may easily reach the water table, and diminishes the role of interflow circulating through the vadose zone. The nature of tides in the Maresme region is semidiurnal, with an amplitude in the order of a few centimeters (~20 – 40 cm). The bathymetry of the Barcelona - Maresme continental shelf is dominated by steps and ridges, with moderate slopes (1.1°), and an average shelf width of 13 km (Durán et al., 2014).

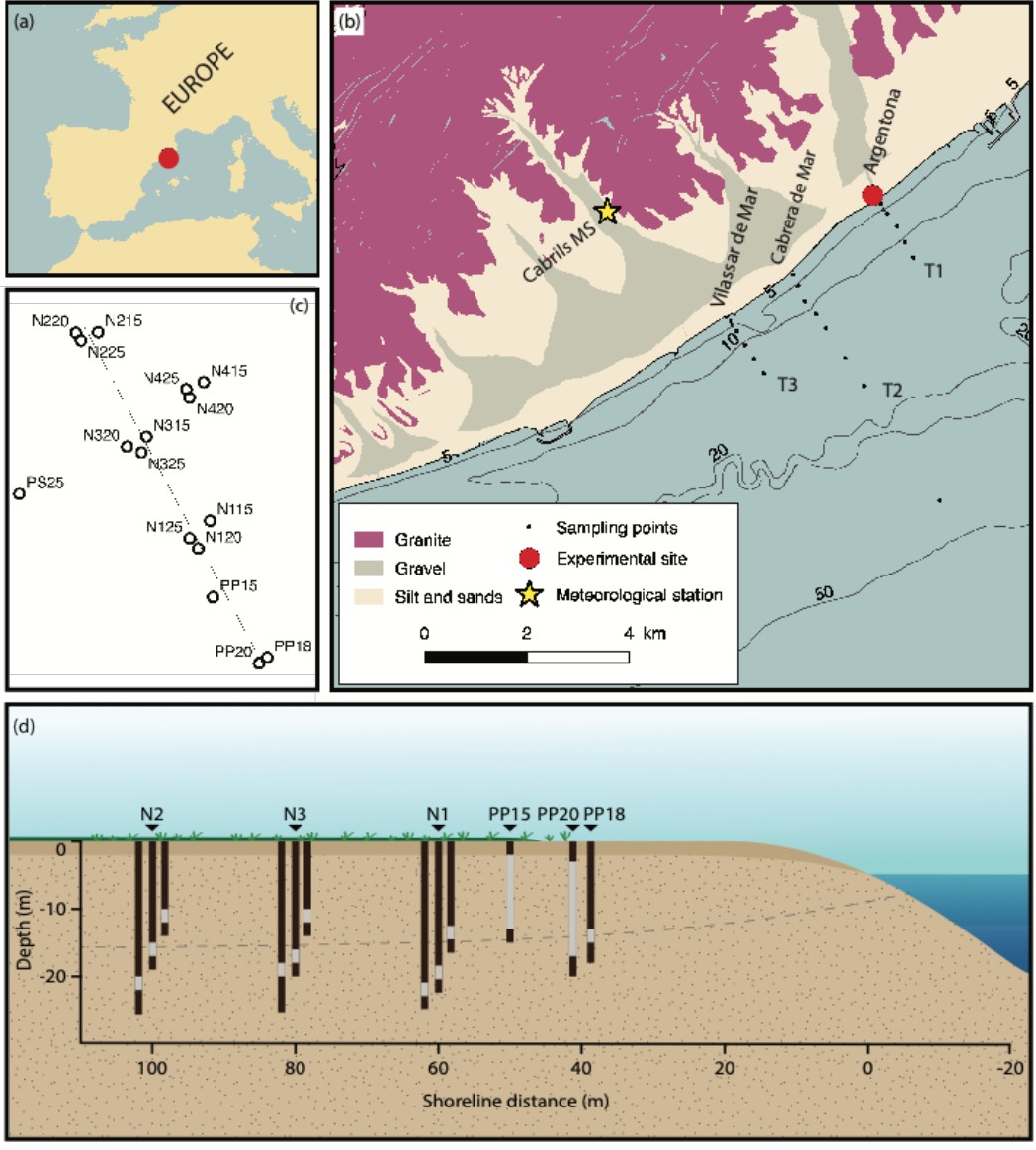

**Figure 1: Study site map. (a) Location map of Maresme County, (b) geological description, bathymetry, and sampling stations, (c) location map of the Argentona site piezometers, and (d) schematic diagram of the piezometers and their screening sections within the perpendicular transect of the experimental site of Argentona. T1, T2, and T3 are the offshore transects associated with the ephemeral streams of Argentona, Cabrera de Mar, and Vilassar de Mar, respectively.**

## 2.2 Field methods

Three samplings were conducted in the southern section of Maresme County during 2019 and 2020. The two first samplings (hereinafter P1 and P2, chronologically) were performed shortly after an EPE with an accumulated precipitation rate of ~90 mm in one day, which corresponds to the 99.6 wet-day percentile (Fig. 2a). The EPE took place on October 22nd, 2019, and P1 and P2 were conducted on October 25th and 29th, 2019 (~4 and 8 days after the rainfall event, respectively). The third sampling (named BF, after "baseflow") was conducted on March 11th, 2020, was not affected by any rainfall event, and was

therefore considered to have been conducted under baseflow conditions (accumulated rainfall of 18 mm in the prior 40 days). Seawater samples were collected at different stations from three perpendicular transects to the coastline corresponding to the ephemeral streams of Argentona, Cabrera de Mar, and Vilassar de Mar (Transects T1, T2, and T3, respectively; Fig. 1). Each transect consisted of 7 offshore stations distributed along the first 1000 m. The central transect (T2) had three additional stations at 1500, 2000, and 4000 m from the coastline. Coastal seawaters were collected directly from the shore by filling 25

L water containers. At each station, surface and deep (only selected stations) seawater samples were collected by placing a submersible pump at ~0.5 m depth and ~1 m above the seabed, respectively. A single runoff water sample was taken at the initial stage of the flood, when the amount of water flowing represented only a thin, 'dirty' layer of water (some cm). Samples were collected for Ra isotopes ($^{223}$Ra, $^{224}$Ra, $^{226}$Ra, and $^{228}$Ra; 25 – 120 L each sample), which are widely applied tracers of SGD (Garcia-Orellana et al., 2021) and nutrient analysis. Depth profiles of salinity and temperature were performed

at each station by using a YSI 600XL probe. Groundwater samples for Ra isotopes (10 – 25 L) and nutrients were collected periodically from 2015 to 2020 in several piezometers at the experimental site of the Medistraes project, located in the coastal alluvial aquifer of the Argentona ephemeral stream (corresponding to Transect 1). The experimental site consists of 16 piezometers located at 30 to 100 m from the coastline with screened depths of 15 to 25 m, with 2 m screened intervals for each (see Diego-Feliu et al., 2021; Folch et al., 2020; Palacios et al., 2019, for more details about the experimental site; Fig. 1).

Each piezometer was purged with a submersible pump to remove at least three times the volume of stagnant water before sampling. Continuous in-situ groundwater level, conductivity, and temperature time-series were monitored at a shallow piezometric well (N3-15; 15 m depth, 2 m screened interval, from 11 to 13 m, 80 m from the coastline) using a CTD diver. Salinity and temperature of groundwater and seawater samples were measured in-situ with two handheld probes (HANNA HI98192 and WTW COND 330I). Rainfall data was obtained from a meteorological station from the Meteorological Catalan

Service (Servei Meteorològic de Catalunya; SMC) at the municipality of Cabrils (see Fig. 1).

## 2.3 Analytical methods

Samples collected for Ra isotopes both in seawater and groundwater were weighted and filtered through $MnO_2$-impregnated acrylic fibers, at a controlled flow rate (< 1 L min$^{-1}$) to ensure the quantitative adsorption of Ra onto the fibers (Moore and Reid, 1973). Fibers were then washed with Ra-free deionized water and partially dried to a fiber-water ratio of 1:1 (Sun and

Torgersen, 1998). Each fiber was measured twice with the Radium Delayed Coincidence Counter (RaDeCC) (Moore and

Arnold, 1996). Short-lived Ra isotopes ($^{223}$Ra, $T_{1/2}$ = 11.4 d; $^{224}$Ra, $T_{1/2}$ = 3.66 d) were quantified using the first RaDeCC measurement. The activities of $^{223}$Ra are not reported in this study because the high $^{224}$Ra activities prevented the proper quantification of $^{223}$Ra with the RaDeCC system, due to cross-talk effect (Diego-Feliu et al., 2020). The second measurement, performed one month after sample collection, was used for quantifying the unsupported activity of $^{224}$Ra (excess $^{224}$Ra; $^{224}$Ra$_{ex}$), by accounting for the activity of its parent, $^{228}$Th, in the fiber. The quantification of $^{224}$Ra was made following the guidelines and limits proposed by Diego-Feliu et al. (2020) in order to avoid interferences inherent to the detection system, while $^{224}$Ra uncertainties were estimated following Garcia-Solsona et al. (2008). The MnO$_2$-fibers were subsequently incinerated, grounded, and transferred to gamma counting vials. After radioactive equilibration (~21 d), the activities of long-lived Ra isotopes ($^{226}$Ra, $T_{1/2}$ = 1,600 y; $^{228}$Ra, $T_{1/2}$ = 5.75 y) were measured using a HPGe gamma spectrometer. The photopeaks of $^{214}$Pb (352 keV) and $^{228}$Ac (911 keV) were used to quantify the activities of $^{226}$Ra and $^{228}$Ra, respectively.

Samples for the analysis of silicate (SiO$_2$), phosphate (PO$_4^{3-}$), nitrite (NO$_2^-$), nitrate (NO$_3^-$), and ammonia (NH$_4^+$) were collected in 10 mL high density polyethylene (HDPE) vials after filtration through nylon syringe filters (pore size: 0.45 μm). Vials were immediately stored in a portable fridge and subsequently frozen at the laboratory until analysis. The analysis was performed using a colorimetric method with an Autoanalyzer AA3 HR (Seal Analytica). The detection limits of the method were 0.016, 0.020, 0.003, 0.006, and 0.003 μM for SiO$_4^{2-}$, PO$_4^{3-}$, NO$_2^-$, NO$_3^-$, and NH$_4^+$, respectively.

## 3 Results

### 3.1 Meteorological and hydrological context

The temporal evolution of groundwater level, conductivity, significant wave height, mean sea level (MSL), and accumulated precipitation from October 2019 to April 2020 are shown in Fig. 2. Three major precipitation events occurred in October 2019 (~90 mm), December 2019 (~100 mm), and January 2020 (~160 mm), which had a direct impact on groundwater level and conductivity. These events are considered extreme precipitation events following the threshold value derived from the 99$^{th}$ wet-day percentile (~75 mm). After each EPE, the groundwater table from the shallow piezometer (N3-15) rose between 60 and 130 cm, gradually recovering the previous values 7 to 10 days after the event. The magnitude in the increase of the groundwater level (60, 70, and 130 cm, respectively) correlated with the amount of accumulated rainfall corresponding to each EPE. Precipitation events were followed by a drastic reduction of groundwater conductivity at the shallow piezometer, reaching minimum values of ~1 mS cm$^{-1}$, and a subsequent gradual increase before the next precipitation event (Fig. 2). As inferred by electric resistivity tomography profiles shown in Palacios et al. (2019), conductivity variations in the piezometric wells of the study site were not derived from dilution with low-conductivity rainwater, but associated with the movement of the mixing zone due to EPE. Significant wave height fluctuations occurred, associated mainly with changes in wind and atmospheric pressure during the EPEs, increasing rapidly from 2 to 5 m above the baseline value of approximately 0.5 m. Similarly, the MSL presented oscillations linked to the EPE, which are usually associated with atmospheric fronts and strong winds, and also

to seasonal meteorology, with higher MSL values from October to December than from January to April. Some hours after the precipitation started, runoff occurred in the ephemeral stream of Argentona, showing a similar patter to that of the reported by previous studies (Cisteró and Camarós, 2014; Martín-Vide, 1985; Riba, 1997). Estimates for runoff discharge or velocity are difficult because in the midd-19th century a set of galleries and dams were constructed at the upper part of the Argentona ephemeral stream (municipality of Dosrius) to collect groundwater and surficial water from this area and transporting it to the municipalities of Barcelona and Mataró. The effect that these structures may have regarding surface runoff is uncertain. However, a soil mass balance (based on type of soil, land use, geology, precipitation, slope, etc.) of the lower part of the Argentona ephemeral stream has been used to provide a semi-quantitative estimate for surface runoff during the rainfall event of October 22nd, 2019. The soil mass balance has been used for a calibrated regional groundwater numerical model of the southern section of Marsme county. The model is not publicly available as it has been developed for a specific work of the Spanish railway public company. According to the soil mass balance, surface runoff associated to this EPE was about 1 hm$^3$.

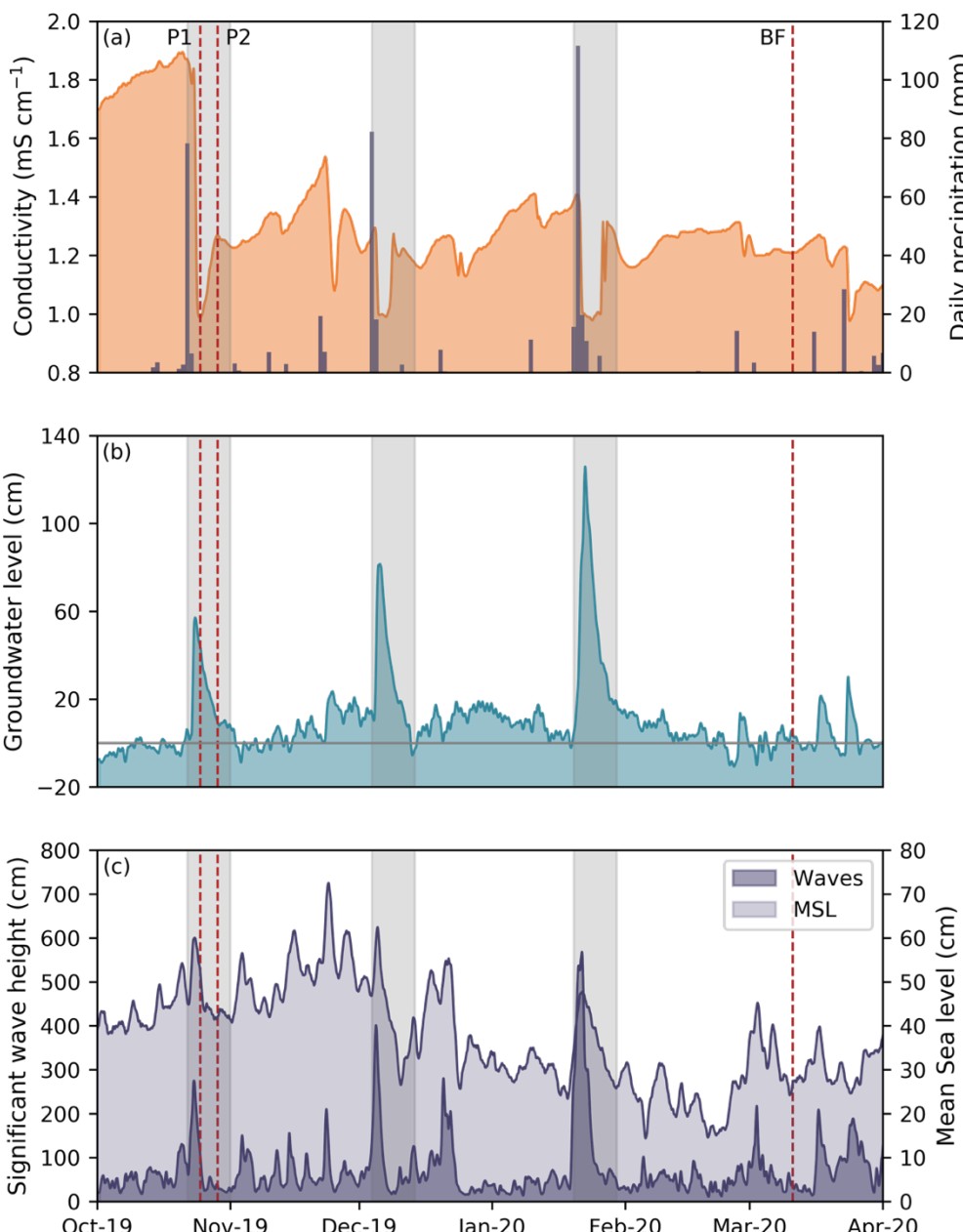

**Figure 2. Temporal evolution of meteorological, oceanographical and hydrogeological data in Maresme County: (a) specific conductivity measured at the shallow piezometer N3-15 and accumulated precipitation (b) groundwater level, and (c) significant wave height and mean sea level. Groundwater level is displayed as the variation of groundwater level relative to the values of March 2020. The data from the buoy and CTD-diver was smoothed by using a low-pass filter (12 h averaged). Red lines indicate the groundwater and seawater samplings performed at the study site (P1, P2, and BF) and grey bands indicate the EPEs that occurred during the monitoring period (10 days after the event are included in the band).**

## 3.2 Radium and nutrient concentrations

The activities of Ra isotopes in groundwater samples measured during the 2015-2020 period in the Argentona site ranged from 10 to 940, 10 to 550, and 1 to 50 Bq m$^{-3}$ for $^{224}$Ra, $^{228}$Ra, and $^{226}$Ra, respectively (see supplementary information (SI); Fig. S1). The activities of Ra increased with groundwater salinity, presenting some variations that are mostly associated with differences in the geological matrix (Beck and Cochran, 2013; Webster et al., 1995). The seawater activities of $^{224}$Ra and $^{228}$Ra isotopes generally decreased with increasing distance offshore for all transects and seawater samplings (Fig. 3). Maximum activities were found in the first sampling after the rainfall event (P1) for $^{224}$Ra (median: 14.4 Bq m$^{-3}$, interquartile range (IQR): 11.7 – 19.9 Bq m$^{-3}$) and $^{228}$Ra (5.8 Bq m$^{-3}$, IQR 4.9 – 7.1 Bq m$^{-3}$). The seawater activities of $^{224}$Ra and $^{228}$Ra in subsequent samplings (P2 and BF) were three to five times lower than those of the sampling after the rainfall event (4.0 Bq m$^{-3}$, IQR 2.4 – 10.2 Bq m$^{-3}$ for $^{224}$Ra; and 1.4 Bq m$^{-3}$, IQR 1.1 – 1.9 Bq m$^{-3}$ for $^{228}$Ra). Seawater activities of $^{226}$Ra were low, and comparable to open seawater activities in all samplings (1.8 Bq m$^{-3}$, IQR 1.4 – 2.0 Bq m$^{-3}$), revealing the lack of major inputs for this Ra isotope. Activities of Ra isotopes in the runoff sample collected during the EPE were 110$\pm$10 Bq m$^{-3}$ for $^{224}$Ra, 90$\pm$5 Bq m$^{-3}$ for $^{228}$Ra, and 18$\pm$1 Bq m$^{-3}$ for $^{226}$Ra. Nutrient concentrations in groundwater from the experimental site of the Argentona ephemeral stream (2015-2020 period) ranged from 10 to 1070 µM for $NO_3^-$, 0.1 to 3.9 µM for $NO_2^-$, 0.3 to 40.1 µM for $NH_4^+$, 0.8 to 10.1 µM for $PO_4^{3-}$, and 50 to 230 µM for $SiO_4^{2-}$ (Fig. S2). Nitrate ($NO_3^-$) and silica ($SiO_4^{2-}$) presented a similar pattern, with maximum concentrations in low salinity samples (Sal < 10) and a downward trend with increasing groundwater salinity. Conversely, the concentrations of nitrite ($NO_2^-$), phosphate ($PO_4^{3-}$), and ammonia ($NH_4^+$) were relatively low for most of the groundwater samples, except for those collected at the shallow piezometers for nitrite, intermediate for phosphate, and deep for ammonia (see SI; Fig. S2).

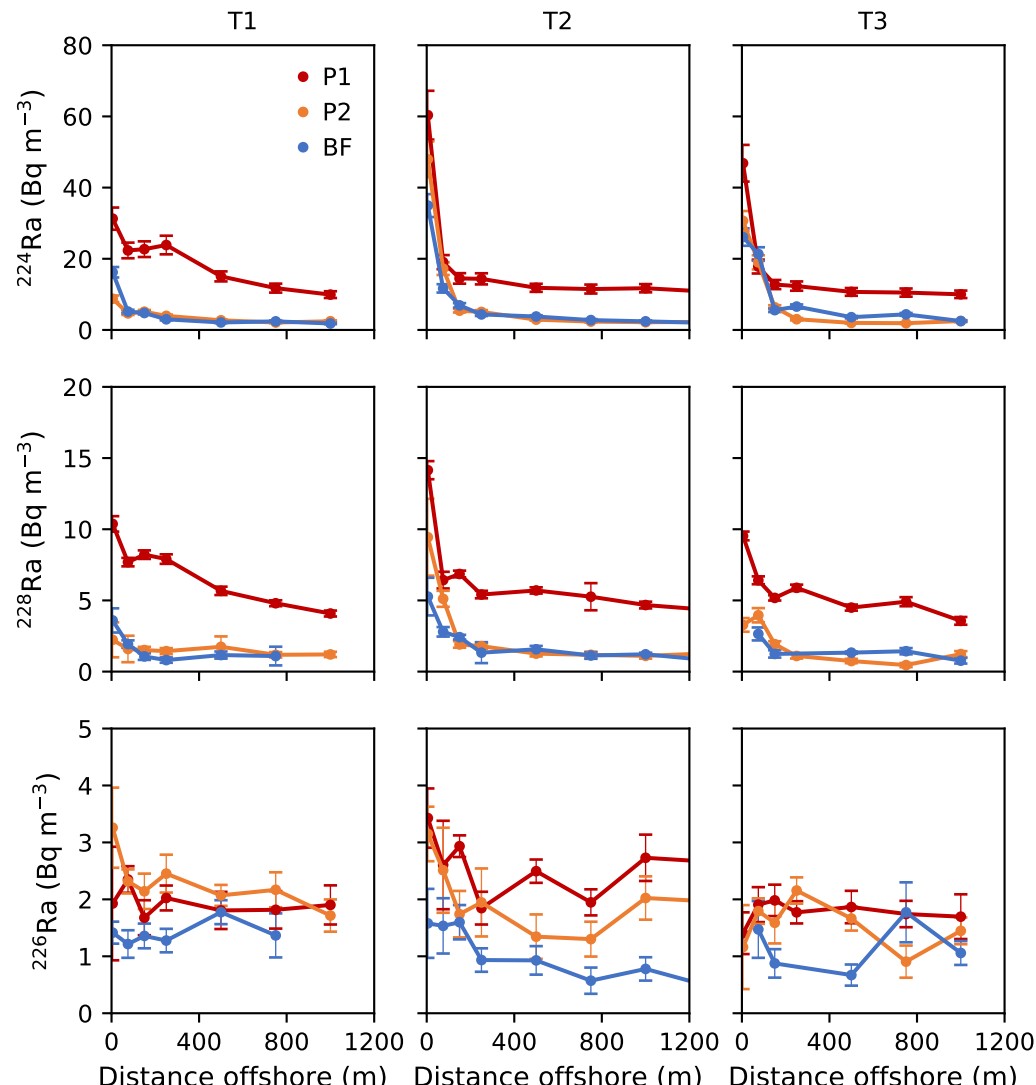

**Figure 3. Radium isotopes activities ($^{224}$Ra, $^{228}$Ra, and $^{226}$Ra) in coastal seawater samples collected during the samplings performed in October 2019 (P1 and P2) and March 2020 (BF) for the three transects perpendicular to the coastline corresponding to the ephemeral streams of Argentona (T1), Cabrera de Mar (T2), and Vilassar de Mar (T3).**

**Discussion**

## 4.1 Groundwater and nutrient fluxes calculation

### 4.1.1 Pathways of submarine groundwater discharge

Submarine Groundwater Discharge incorporates a set of water flow processes involving the discharge of fresh groundwater and the circulation of seawater through permeable sediments (Garcia-Orellana et al., 2021; Michael et al., 2011; Santos et al., 2012). The driving forces and pathways of these processes likely determine the extent of the chemical reactions occurring in the subterranean estuary (Moore, 1999). Therefore, considering all the different SGD pathways concurrently occurring in a specific study site is fundamental for deriving reliable estimates of SGD and associated nutrient fluxes (Garcia-Orellana et al., 2021). The characteristics of coastal alluvial aquifers linked to the presence of ephemeral streams in the Maresme county may favor the concurrent occurrence of different water flow processes. Indeed, previous works at the study site of Argentona have already shown that different SGD components coexist (Diego-Feliu et al., 2021; Folch et al., 2020; Martínez-Pérez et al., 2022; Palacios et al., 2019). Meteoric groundwater flowing seawards and recirculated seawater mix in multiple aquifer levels which are separated by semi-confining thin layers of silt and clays. The different aquifer units and mixing zones may promote the combined discharge of fresh and saline groundwater (brackish) at the coastline or far beyond depending on the continuity of the confining layers (Folch et al., 2020; Martínez-Pérez et al., 2022). A seawater recirculation cell has been observed in the upper part of the aquifer (i.e., upper saline plume; Robinson et al., 2018), where seawater infiltrates through the most surficial layers due to wave set up and/or sea level variations associated to extreme precipitation events (EPEs) or storm surges (Palacios et al., 2019). Offshore exchange of seawater due to the movement of the freshwater-saltwater interface may also occur in response to the increased infiltration of rainwater inland associated with EPEs. Interflow may occur after an EPE, however it may probably reach easily the water table due to the thin vadose zone (2 to 3 m) and the high permeability of the surficial materials (Martínez-Pérez et al., 2022). Porewater exchange may also occur due to its almost ubiquitous character and the disparate mechanisms driving the water flow (Santos et al., 2012).

In this work, for convenience the water flow processes described above have been clustered into two main SGD components; brackish and saline SGD. Brackish SGD is defined here as the combined discharge of meteoric groundwater and density-driven (long-term) recirculation of seawater through the saltwater wedge regardless of the degree of mixing between the two water masses and the aquifer unit considered. It should be noticed that this SGD component (1) does not represent a net water input to the coastal ocean, (2) exclude water flow processes solely involving the recirculation of seawater through permeable sediments, and (3) also include the contribution that interflow may have on groundwater discharge after the occurrence of an EPE. On the other hand, beach-face recirculation of seawater through the upper saline plume, porewater exchange, and offshore exchange of seawater due to the movement of the freshwater-saltwater interface are ascribed here to the Saline SGD component. This SGD component (1) represents a net zero water input to the coastal ocean for timescales longer than that of the process driving its oscillations, and (2) comprises a set of water flow processes with disparate spatiotemporal scales (minutes to days). Besides the obvious difference between these two components in terms of water composition and origin, it

should be also noticed that brackish SGD may be mediated by medium to long-term term spatiotemporal scale processes and conversely, saline SGD is likely to be governed by short to medium spatiotemporal scale flow paths.

The different spatiotemporal scales of both SGD components are especially important when using Ra isotopes as tracers of SGD. In fact, these isotopes are instrumental for differentiating SGD pathways, since their enrichment rates strongly depend on the transit time of groundwater through the coastal aquifer (e.g., Diego-Feliu et al., 2021; Michael et al., 2011). Coastal seawater samples collected during the three samplings performed in Maresme County were enriched in both $^{224}$Ra and $^{228}$Ra relative to offshore waters (Fig. 3), suggesting the occurrence of a land-based Ra source. Whilst the enrichment in $^{224}$Ra may result from any groundwater discharge, regardless of spatiotemporal scale, due to its short half-life ($^{224}$Ra is enriched in all SGD pathways), coastal waters enriched in $^{228}$Ra may be indicative of long-scale SGD pathways (e.g., terrestrial groundwater discharge; Rodellas et al., 2017; Tamborski et al., 2017a).

The activity ratio (AR) of $^{224}$Ra/$^{228}$Ra can similarly be used to evaluate the temporal scale of SGD pathways (Diego-Feliu et al., 2021). Thus, the $^{224}$Ra/$^{228}$Ra AR found in coastal seawater samples after the EPE from October 2019 decreased from a baseline value of 6 to approximately 4. This decrease is simultaneously followed by an increase in absolute $^{228}$Ra activities, which are two-times higher than those in baseflow conditions (Fig. 3). Although many processes may be explanatory of the activities found in coastal waters, the observed trends for $^{224}$Ra/$^{228}$Ra AR and $^{228}$Ra activities can indicate that the relative contribution of the brackish component of SGD, which is characterized by $^{224}$Ra/$^{228}$Ra ARs close to the equilibrium value (1.0 to 2.2; Diego-Feliu et al., 2021), increased during the occurrence of the EPE. This is also coherent with the increase in groundwater level (Fig. 2), and therefore hydraulic forcing, after the EPE of October 2019.

### 4.1.2 Submarine groundwater discharge

Submarine groundwater discharge associated with brackish and saline water flows were determined by means of a steady-state mass balance of $^{224}$Ra and $^{228}$Ra. The assumptions, considerations, and models used to estimate SGD are discussed in detail in appendix A. The sampling influenced by the EPE of October 2019 (P1) presented the highest total SGD (brackish and saline) (510 ·10$^3$ m$^3$ km$^{-1}$ d$^{-1}$, IQR: 320 – 890 ·10$^3$ m$^3$ km$^{-1}$ d$^{-1}$), one order of magnitude higher than those flows from subsequent samplings: 40 ·10$^3$ m$^3$ km$^{-1}$ d$^{-1}$ (IQR: 30 – 90 ·10$^3$ m$^3$ km$^{-1}$ d$^{-1}$) for P2 and 70 ·10$^3$ m$^3$ km$^{-1}$ d$^{-1}$ (IQR: 40 – 110 ·10$^3$ m$^3$ km$^{-1}$ d$^{-1}$) for BF (Fig. 4). Thus, the effect of this EPE on the discharge of groundwater lasts only for a few days, as the SGD estimates of the second sampling (P2; 8 days after the event) are comparable to those from baseflow conditions. Besides the similitudes in SGD estimates from these two samplings (P2 and BF), they also present similar groundwater levels, conductivities, and Darcy's flow estimates (Fig. 2 and Fig. B1). Thus, the temporal extent of the EPE effects on SGD is consistent with the recovery of groundwater level, which commonly occurs from 7 to 10 days after the rainfall ceases (Fig. 2).

In baseflow conditions, the brackish component of SGD (including fresh groundwater and density-driven seawater discharge) represented 60% of the total SGD (Fig. 4). The relative contribution of this SGD component increased after the rainfall event of October 2019 to up to 75% of the total SGD. This is consistent with the variation on the $^{224}$Ra/$^{228}$Ra AR in coastal seawater after the EPE (see Section 4.1.1) and coherent with Darcy's flow calculations (Appendix B). These estimates of the relative

255      contribution of the brackish component are generally much larger than estimates of fresh groundwater discharge for the
Mediterranean Sea (1 - 25%, Rodellas et al., 2015), global estimates (10%, Kwon et al., 2014; 0.06%, Luijendijk et al., 2020),
and local studies (5 - 55%; Alorda-Kleinglass et al., 2019; Beck et al., 2008; Kiro et al., 2014; Knee et al., 2016; Rodellas et
al., 2017; Tamborski et al., 2017a). This difference most likely emphasizes that whereas the studies presented above are mainly
focused on distinguishing fresh and saline SGD, here we are targeting brackish (encompassing meteoric groundwater and
260      recirculated seawater) and saline SGD, as previously discussed. It should also be noticed that estimates presented in this study
should be taking as semi-quantitative in view of the biases, limitations, and uncertainties discussed in detail in appendix A
(e.g., endmember selection, steady-state assumption, lack of consideration for runoff). However, these limitations are inherent
to almost any SGD study and especially those using Ra isotopes as tracers (Garcia-Orellana et al., 2021; Rodellas et al., 2021),
and may not invalidate the implications derived from this study which is trying to ascribe the relative significance of EPEs in
265      water and nutrient fluxes to the coastal ocean.

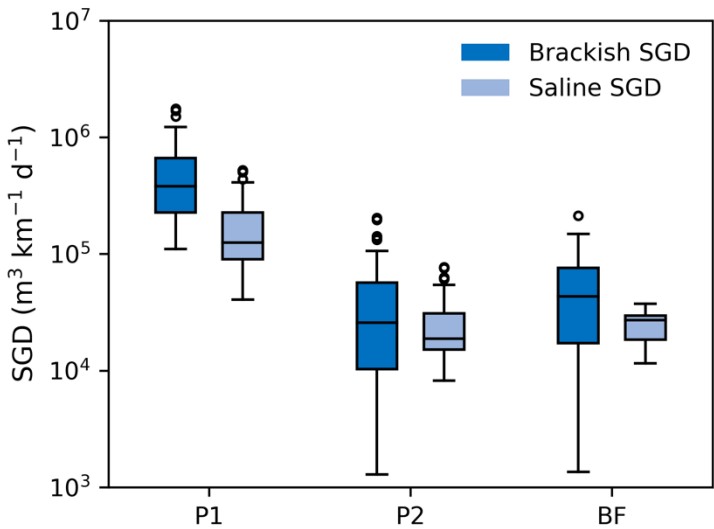

**Figure 4. Coastal-normalized flow of brackish and saline submarine groundwater discharge (dark and light blue, respectively) for the three samplings at Maresme County.**

### 4.1.3 SGD-driven nutrient fluxes

SGD-driven nutrient fluxes were estimated by considering the Ra-derived flows of brackish and saline SGD and the respective nutrient concentration in groundwater from both fractions (see Appendix A). Notice that results are not expressed here in terms of net nutrient inputs since the fluxes of nutrients from the coastal ocean to the coastal aquifer are not considered. However,
270      the influence that these fluxes have for the calculations may be negligible since concentrations of nutrients in seawater are orders of magnitude lower than those in groundwater (see SI; Fig. S2). Total SGD-driven fluxes in baseflow conditions for dissolved inorganic nitrogen (DIN), dissolved inorganic phosphorus (DIP), and dissolved silicate (DSi) derived from median

SGD estimates in the study site were 16.2, 0.06, and $5.4 \cdot 10^3$ mol km$^{-1}$ d$^{-1}$, respectively (Fig. 5). The median fluxes, normalized by the study site area, were lower compared with median SGD-derived nutrient fluxes estimated worldwide for DIP and DSi, but significantly higher for DIN (2.7 times higher; Santos et al., 2021). The DIN:DIP ratio was 390:1, much higher than the Redfield ratio of 16:1, but comparable with SGD-derived input in the Mediterranean Sea (80:1-430:1; Rodellas et al., 2015) and in studies worldwide (259±1090:1; Santos et al., 2021). The high loads of N and the disproportionate ratio DIN:DIP relative to the Readfield ratio in the study site may result from the lixiviation of nitrogen from agricultural activities (representing ~15% of the total land-use; Rufí-Salís et al., 2019), and the attenuation of P along groundwater flow paths due to adsorption onto Mn/Fe oxides present in the coastal aquifer (Robinson et al., 2018; Spiteri et al., 2007, 2008b).

After an EPE, the supply of all nutrients increased due to the higher brackish and saline SGD associated with these episodes (Fig. 5). Fluxes after the EPE (P1) were 9 times higher for DIN and 7 times higher for DIP and DSi than those in baseflow conditions. The predominant pathway for DIN, DIP, and DSi discharge to the coastal ocean was the brackish component of SGD. This pathway represented ~60% of the total inputs of DIP and DSi in baseflow conditions and up to ~75% after an EPE (Fig. 5). Nitrogen inputs during EPE and in baseflow conditions were chiefly governed by the discharge of brackish SGD (~99% of the total DIN inputs; Fig. 5). The significant difference between the supply of nitrogen through brackish and saline SGD relies on the high concentrations of nitrate (~1,000 µM) in coastal aquifer freshwater (see SI; Fig. S2), which exceed the maximum groundwater concentration for drinking water set by the World Health Organization (WHO, 2011). Contrastingly, saline SGD is a relevant source of nitrite and ammonia, representing ~70% and ~40% of the total fluxes, respectively. It should be noticed that nutrient fluxes were estimated by multiplying the volumetric water flux of brackish and saline SGD by the minimum nutrient concentration from a set of onshore samples, selected following the criteria used for the Ra endmembers, as explained in the appendices (see appendix A.2.4). Since it was not possible to directly collect the discharging groundwater, by using onshore samples we are implicitly assuming that no nutrient transformation occurred between the sampling point and the discharge point, within the subterranean estuary (Cook et al., 2018). This assumption is perhaps one of the main sources of uncertainty in the reported nutrient fluxes as it has already shown by many authors (Sawyer et al., 2014; Weinstein et al., 2011; Wong et al., 2020). It should also be noted that these SGD-derived nutrient estimates may be biased due to the groundwater endmember selection, since nutrient concentrations in discharging groundwaters may vary during EPE due to dilution, increasing lixiviation of fertilizers, or enhancement of biogeochemical reactions in the mixing zone of coastal aquifers (Spiteri et al., 2008a). Although all the assumptions made for nutrient fluxes quantification may result in high degrees of uncertainty, the results presented in this study enable the assessment of EPE significance as a major driving force transporting nutrients to the coastal ocean.

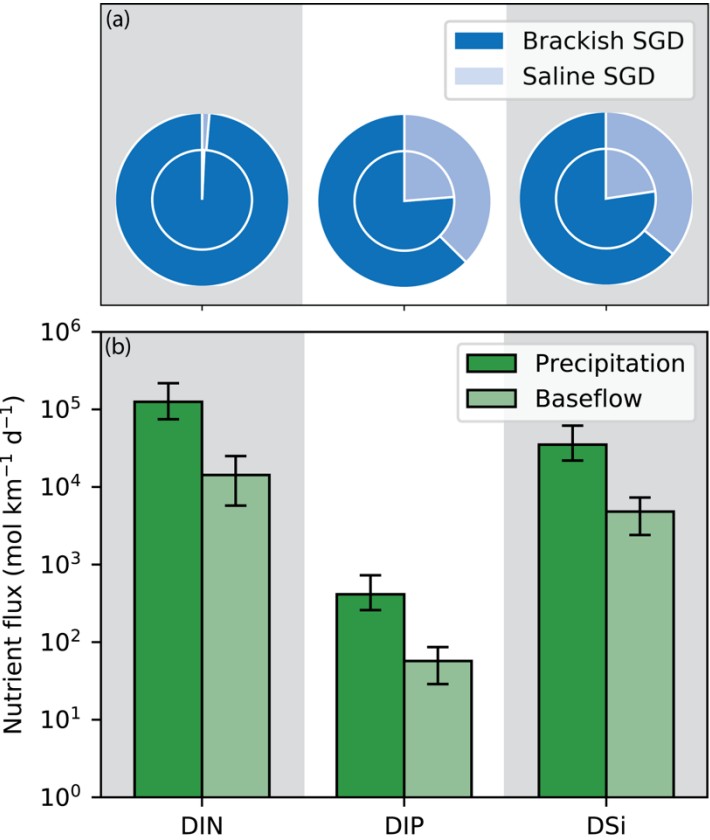

**Figure 5.** SGD-derived nutrient fluxes of dissolved inorganic nitrogen, dissolved inorganic phosphorus, and dissolved silicate at Maresme County. (a) relative contribution of brackish (dark blue) and saline (light blue) SGD during EPE (inner pie chart) and in baseflow conditions (outer pie chart). (b) Median nutrient fluxes at Maresme County during the EPE of October 2019 (P1) and in baseflow conditions (BF) (fluxes normalized by the coastline length). Error bars indicate the interquartile range (25% and 75% percentile).

## 4.2 Implications of EPE on SGD and associated nutrient fluxes

### 4.2.1 Episodic events

Although several studies have focused on understanding the seasonal dynamics of SGD (Charette, 2007; Gwak et al., 2014;
Michael et al., 2005; Rodellas et al., 2017), limited research has been done on SGD driven by episodic events (Adyasari et al.,
2021; Wilson et al., 2011). This is mainly because of the inherent difficulties related to monitoring and sampling during and
after these extreme events. Some studies have already shown that SGD may vary in direct or delayed response to
meteorological and oceanic episodic events such as sea-level anomalies (Gonneea et al., 2013), waves (Bakhtyar et al., 2012;
Rodellas et al., 2020; Sawyer et al., 2013), hurricanes (Hu et al., 2006), typhoons (Cho et al., 2021; Sugimoto et al., 2016), and
temperature inversion (Moore and Wilson, 2005). Regarding SGD induced by precipitation events: Sugimoto et al. (2016)
reported high values of SGD eleven days after a precipitation event in Obama Bay (Japan); Gwak et al. (2014) suggested that

SGD in the II-Gwang watershed (South Korea) was partially triggered by intensive precipitation events; and Uddameri et al. (2014) indicated that precipitation events associated with the Emili hurricane contributed to the SGD in Baffin Bay (USA). At our study site, the significant increase (1 order of magnitude; Fig. 4) in both the brackish and saline SGD after EPE, may be mediated by different processes: (1) increase of fresh groundwater discharge due to aquifer infiltration of rainfall, subsequent increase of hydraulic gradient, and displacement of stored water towards the sea (Anwar et al., 2014; Palacios et al., 2019; Santos et al., 2012; Yu et al., 2017), (2) increase in the exchange of seawater and density-driven discharge due to movements of the fresh-saltwater mixing zone, and (3) increase of shoreface circulation of seawater and porewater exchange due to the effect of sea level rise and waves associated with the EPE (Fig. 2).

The higher total SGD driven by EPE, especially the increase of brackish relative to saline SGD during these episodic events (Fig. 4), induces the transport of large amounts of nutrients from the freshwater fraction of coastal aquifers into the coastal ocean. These fluxes, which are substantially higher than those in baseflow conditions (Fig. 5), may represent a significant periodic and episode-related nutrient input of particular relevance in sites where surface water renewal is limited (e.g., coastal lagoons and/or semi-enclosed bays), EPE occurs frequently (e.g., Mediterranean region), and the response to EPE is fast due to the geological and geomorphological characteristics of the coastline (e.g., alluvial aquifers in ephemeral stream-dominated areas, such as Maresme County). However, in other areas the response to EPEs may be much slower. For instance, aquifers with a high thickness of vadose zones, confined aquifers with recharge areas far from the coastal zones, or systems with soils with water deficits (among other factors), may smooth or delay the effects of EPEs. The biological and ecological implications of these events- which may include eutrophication, formation of red and green tides, and mass fish death (Hu et al., 2006; Lee et al., 2010; Montiel et al., 2019; Valiela et al., 1990; Zhao et al., 2021)- are far from being understood, and require further attention.

### 4.2.2 Annual SGD estimates

A proper understanding of the temporal patterns of SGD in local to regional-scale studies is essential for deriving annual estimates for predicting reliable ocean budgets of nutrients and other dissolved compounds (Luijendijk et al., 2020; Santos et al., 2021). However, most SGD studies are conducted in periods with stable meteorological conditions to evaluate baseflow SGD and associated nutrient, metal, or contaminant fluxes. Based on the results obtained from monitoring the EPE occurred at Maresme County in October 2019, the discharge of groundwater associated to this single event accounted for 13% (IQR: 5 – 40%) and 8% (IQR: 5 – 18%) of the annual brackish and saline fraction of SGD, respectively. Moreover, the nutrient inputs resulting from this event (lasting only 8 days; ~2% of the year) represented 13% (IQR: 5 – 40%) for DIN, and 11% (IQR: 5 – 30) for DIP and DSi, of the yearly supply of nutrients at the study site. The increase in SGD-driven nutrient fluxes during these events may be mediated, on the one hand by the total SGD increase, but also because this increase is more significant for brackish SGD, which presents higher concentrations of nutrients (see supplementary information; Fig. S2), relative to saline SGD. These results suggest that annual estimates based on samplings conducted in baseflow conditions may systematically underestimate SGD and associated nutrients, particularly at study sites affected by EPEs or other episodic events that can

significantly impact SGD. Periodic and seasonal samplings may be taken as snapshots, and only representative of the time periods with similar environmental conditions. A better characterization of the hydrological and meteorological context is necessary in pursuit of more reliable annual estimates, which may include seasonal and episodic-related variations. In this scenario, alternative methods such as groundwater level monitoring, Darcy's law calculations, electric resistivity tomography characterization, among other methods, may be instrumental in the design of proper sampling strategies and to capture seasonal and episodic variations (e.g., Folch et al., 2020; Palacios et al., 2019).

### 4.2.3 Climate change

Climate change and its associated social and environmental impacts have become one of the most pressing scientific challenges for the 21st century. This requires the acquisition of a holistic and integrative knowledge of systems and processes for modelling and predicting future scenarios. Fundamental research relating environmental key variables (e.g., temperature, sea-level rise, precipitation) with social processes (e.g., land demand, coastal overpopulation, groundwater squeeze) becomes crucial to this aim. Research on SGD is no exception to this trend, and in recent decades, several studies have evaluated the key factors contributing to groundwater flows discharging into the ocean. Some examples of SGD research linked to climate change include: understanding sea-level and/or tidal controls on SGD (Gonneea et al., 2013; Wilson et al., 2015); the influence of land-use changes (Rufí-Salís et al., 2019); the relationship to increasing seawater intrusion (Werner et al., 2013); and the fate and evolution of nutrients in groundwater (Beusen et al., 2013; Van Meter et al., 2018; Tait et al., 2014).

The precipitation-recharge relationship is one of the key parameters influencing the discharge of groundwater. Indeed, the amount of precipitation and the frequency and/or distribution of rainfall events, together with the hydrogeological characteristics of the receiving aquifers, strongly affects both the quantity and chemical quality of groundwater discharge to the ocean (Kundzewicz and Döll, 2009; Stigter et al., 2014). Climate models indicate substantial spatial variation in future changes to average precipitation. Whilst the increased specific humidity and transport of water vapor from tropic regions is likely to increase the amount of precipitation in high latitude land masses, precipitation in subtropical and semi-arid regions, like the Mediterranean Sea, is expected to decrease. Simultaneously, EPE in these regions are likely to increase in intensity and frequency (IPCC, 2021). Consequently, the yearly recharge of groundwater associated with precipitation in the Mediterranean region may diminish, also reducing the annual discharge of groundwater. In that scenario, EPEs may become a major driving force of SGD, having a dominant significance in the annual fluxes of solutes to the coastal ocean.

The potential relevance of EPEs on SGD in future conditions can be qualitatively evaluated for Maresme County by considering the period from October 2019 to April 2020, when 3 precipitation events of >75 mm occurred (Fig. 2). Since the historical recurrence of EPE in the area is around 13 months (based on the meteorological data from 2015 to 2020), this 7-month period can be considered as a future-like year with increased recurrence of EPE. Assuming that each one of the EPE produces an increase in SGD comparable to the event monitored in this study, the relative contribution of SGD during EPEs would represent 30% (IQR: 15 – 70%) and 22% (15 – 40%) of the annual SGD issued by the brackish and saline fraction, respectively. Similarly, nutrient fluxes associated with EPE for this period would represent 34% (IQR: 15 – 70%) for DIN and

30% (IQR: 15 – 60%) for DIP and DSi of the yearly nutrient inputs supplied by SGD. Notice that due to the assumptions made in the determination of groundwater and nutrient fluxes (e.g., steady state, endmember selection; see appendix A), the estimates are seemingly conservative, especially considering that the monitored EPE is minor relative to other EPEs occurring in 2019 and 2020 (Fig. 2). These estimates emphasize the need for integrating episodic events, such as EPE, in future climate change scenarios, in order to properly constrain the fluxes of groundwater and solutes to the coastal ocean driven by SGD.

## 5 Conclusions

Extreme precipitation events are potential drivers of submarine groundwater discharge and driven nutrients to the coastal ocean. The lack of studies assessing the impact of these episodic events mask the implications that they may have for coastal geochemical cycles, coastal ecosystems, nutrient budgets, and hydrological cycle estimates. We have assessed the fluxes of SGD induced by an EPE in an ephemeral stream-dominated basin of the western Mediterranean Sea. SGD induced by the EPE increased by one order of magnitude and represented up to 13 and 8% of the total annual discharge of groundwater of brackish and saline SGD, respectively. Similarly, fluxes of nutrients driven by SGD during an EPE represented 11 - 13% of the annual total SGD, and up to ~30% during abnormally rainy seasons. This study highlights the relevance of these extreme events on the discharge of groundwater and solutes to the coastal ocean, noting their implications for annual SGD estimates and the possible consequences on coastal biogeochemistry cycles. The results of this study contribute to the understanding of the evolution of SGD with respect to future climate change scenarios, presenting an opportunity for streamlining future research in order to help managers and policy makers better estimate SGD and its related consequences.

## 6 Appendices

### A SGD and nutrient fluxes calculations

In this appendix, we develop the methodology used for determining the SGD and associated nutrient fluxes to the coastal ocean. This includes the definition of the conceptual model and the discussion of the model assumptions used in the calculations.

### A.1 Radium mass balance

The magnitude of SGD and its associated nutrient fluxes are quantified in this study by using Ra isotopes, which is one of the most commonly applied techniques (Garcia-Orellana et al., 2021; Taniguchi et al., 2019). Whilst single isotopes can be used for quantifying SGD driven by single pathways, the combination of different isotopes is instrumental in discriminating SGD in sites with multiple pathways (Alorda-Kleinglass et al., 2019; Charette, 2007; Rodellas et al., 2017; Tamborski et al., 2017b). In this study, Ra isotopes are used for discriminating and quantifying both the fluxes of brackish and saline SGD. Whilst brackish SGD, as defined in this study is mainly associated to long-term SGD processes (meteoric groundwater discharge and

density-driven seawater recirculation over the saltwater wedge), saline SGD comprises disparate discharge short-term processes solely involving the circulation of seawater through permeable sediments or the coastal aquifer (i.e., porewater exchange, shoreface circulation of seawater, seasonal exchange of seawater; Garcia-Orellana et al., 2021; Michael et al., 2011).

A steady-state mass balance of short-lived $^{224}$Ra ($T_{1/2}$ = 3.66 d) and long-lived $^{228}$Ra ($T_{1/2}$ = 5.75 y) isotopes was constructed as follows:

$$F_{BSGD} \cdot {}^{228}Ra_{BSGD} + F_{SSGD} \cdot {}^{228}Ra_{SSGD} = \frac{I_{ex-Ra228}}{T_F} + I_{Ra228} \cdot \lambda_{Ra228}$$
$$F_{BSGD} \cdot {}^{224}Ra_{BSGD} + F_{SSGD} \cdot {}^{224}Ra_{SSGD} = \frac{I_{ex-Ra224}}{T_F} + I_{Ra224} \cdot \lambda_{Ra224}$$

(A1)

The two terms on the right-hand side of Eq. (A1) represent the Ra outputs and include the offshore exchange of Ra, due to the mixing between coastal and open ocean waters, and Ra decay. The terms $I$ and $I_{ex}$ refer to the mean inventories [Bq m$^{-1}$] of

Ra and excess Ra (inventory of excess Ra concentrations in coastal waters relative to open ocean), respectively, $T_F$ is the Ra flushing time [d], and $\lambda$ is the Ra decay constant [d$^{-1}$]. The mean Ra inventories were determined by averaging the activity (or excess activity) [Bq m$^{-3}$] of Ra at each station normalized by depth and distance to the shore. The normalization was made by integrating the rectangular trapezoid area confined within the distance between two subsequent stations and their respective depth. The two terms on the right-hand-side of Eq. (A1) account for Ra inputs to the study site, which are supplied via brackish

and saline SGD (total flux of Ra; $F_{Ra}$ [Bq m$^{-1}$ d$^{-1}$]). The activities [Bq m$^{-3}$] of Ra in the discharging groundwater ($Ra_{BGD}$ and $Ra_{SSGD}$; Ra endmember) were used to convert the Ra flux of both isotopes concurrently, to a coastline-normalized volumetric flow [m$^3$ km$^{-1}$ d$^{-1}$] for each of the SGD pathways (Brackish SGD, $F_{BSGD}$ and Saline SGD, $F_{SSGD}$). The terms and values used for the Ra mass balance for each sampling are shown in Table A1.

**Table A1. Definition of terms, values and units used for the Ra mass balance for each sampling (P1, P2 and BF). Data in brackets represent the interquartile range (1st and 3rd quartile).**

| Term | Definition | Values | | | Units |
|---|---|---|---|---|---|
| | | P1 | P2 | BF | |
| $I_{Ra224}$ | $^{224}$Ra inventory | 113±15 | 26±2 | 31±9 | ·10$^3$ Bq·m$^{-1}$ |
| $I_{ex-Ra224}$ | $^{224}$Ra excess inventory | 98±15 | 23±2 | 28±9 | ·10$^3$ Bq·m$^{-1}$ |
| $I_{Ra228}$ | $^{228}$Ra inventory | 46±3 | 11±2 | 9±3 | ·10$^3$ Bq·m$^{-1}$ |
| $I_{ex-Ra228}$ | $^{228}$Ra excess inventory | 41±3 | 10±2 | 8±3 | ·10$^3$ Bq·m$^{-1}$ |
| $F_{Ra-224}$[*1] | $^{224}$Ra flux | 62±9 | 9±1 | 11±4 | ·10$^3$ Bq·m$^{-1}$·d$^{-1}$ |
| $F_{Ra-228}$[*1] | $^{228}$Ra flux | 16.8±1.3 | 1.8±0.4 | 1.6±0.5 | ·10$^3$ Bq·m$^{-1}$·d$^{-1}$ |

| | | | | | |
|---|---|---|---|---|---|
| $F_{Ra-224}/F_{Ra-228}$ | $^{224}$Ra/$^{228}$Ra flux ratio | 3.7±0.3 | 5.2±1.2 | 7.6±2.7 | - |
| $T_F$*2 | Ra flushing time | 2.4±0.9 | 5.6±1.9 | 5.0±1.7 | d |
| ***Unknowns*** | | | | | |
| $F_{BSGD}$ | Brackish SGD flow | 380 (235 - 660) | 25 (10 - 55) | 45 (15 - 75) | ·10$^3$ m$^3$·km$^{-1}$·d$^{-1}$ |
| $F_{SSGD}$ | Saline SGD flow | 125 (90 - 225) | 20 (15 - 30) | 25 (20 - 30) | ·10$^3$ m$^3$·km$^{-1}$·d$^{-1}$ |

*[1] Flux of Ra supplied by brackish and saline SGD, left-hand side of Eq. (A1)
*[2] Radium fluxing time determined as water apparent age following Moore (2000)

### A.2 Model assumptions and considerations

**A.2.1 Sources and sinks of Ra**

The proposed model for quantifying the Ra flux to the sea assumes that brackish and saline groundwater discharge are the only sources of Ra at the study site (Eq. A1). Diffusive fluxes of Ra from sediments were considered negligible, due to the presence of coarse-grained sands with low specific surface area (Luek and Beck, 2014) and are assumed to represent low (10%) inputs compared to the total Ra inputs (Beck et al., 2007; Garcia-Orellana et al., 2014, 2021). Atmospheric deposition was discarded

as a major source of Ra since its contribution in small-scale study sites is often <<1% (Garcia-Orellana et al., 2021). Production of Ra from dissolved Th was implicitly included by reporting the activities of Ra isotopes as 'excess' Ra activities (activities non-supported by their progenitors). Ra inputs from surface water were also discarded for the sampling conducted in March 2020 (BF) due to the total absence of runoff during the sampling period. In October 2019, 4 days before the first sampling conducted at the study site, runoff occurred in direct response to an EPE (~90 mm). However, considering the flushing time

of Ra isotopes in the coastal system (see S1.2.2), the Ra delivered by this punctual runoff may have decreased by >90% for the first sampling (P1) and by >99% for the second sampling (P2), due to decay and mixing with offshore waters. Yet, the relative contribution that runoff may have had on Ra inventories during the two first samplings were calculated using the Ra activities of the runoff sample collected during the EPE, and the calculated runoff discharge by using soil mass balance. Although the estimates may be uncertain, the results indicate that the relative significance of runoff derived from the EPE in

the Ra inventories are 2 and 1% for $^{224}$Ra, and 1 and 8% for $^{228}$Ra at the first and second sampling, respectively. Notice that these values are low and comparable with the common uncertainties derived from the measurement of Ra isotopes. It should be also noticed that the calculated Ra inputs through surface runoff are likely overestimated since the surface water sample was collected at the beginning of the flood and it is representative only of the initial thin and 'dirty' water (with more particles per mass of water) flow and not of the 'cleaner' water mass which represents most of the total runoff discharge. Therefore, Ra

runoff was discarded as a major source of Ra isotopes to the coastal ocean during the sampling periods. The decay of Ra and

the exchange with offshore waters were considered as the major sinks of Ra. The decay was assessed via the Ra inventories at in the study site and the offshore exchange, by evaluating the flushing time of Ra ($T_F$).

### A.2.2 Radium flushing time

The flushing time of Ra ($T_F$) is a parameter that describes the transport of Ra in surface water bodies due to advection and dispersion processes (Monsen et al., 2002). In Ra mass balances, this parameter is fundamental for evaluating the exchange of Ra between coastal and offshore waters. In this work, rather than evaluating Ra flushing times, we used [224]Ra/[228]Ra of coastal and offshore waters (1,000 m from coastline) to determine the water apparent age ($T_W$) (Moore, 2000). The water apparent age is a good proxy for temporal scales of advective and mixing processes occurring at the study site. Coastal waters in Maresme County presented [224]Ra/[228]Ra activity ratios ranging from 1.6 to 2.9 times higher than those of offshore waters, which led to seawater apparent ages of 2.4±0.9, 5.6±1.9, and 5.0±1.7 days for the first, second, and third sampling, respectively. The lower seawater residence time of the first sampling is coherent with the oceanographic conditions (e.g., higher winds, waves and currents that enhanced advection and exchange with offshore waters) linked to the extreme precipitation event occurring 4 days before (see Fig. 2).

### A.2.3 Steady state conditions

Steady state conditions (i.e., tracer inventories do not vary with time; $da/(dt \cdot V) = 0$) are often assumed in Ra mass balances (e.g., Alorda-Kleinglass et al., 2019; Beck et al., 2008; Rodellas et al., 2017). This assumption implies that Ra inputs and outputs are balanced for a time period equivalent to the tracer residence time in the system (Rodellas et al., 2021). In Maresme County, the tracer residence time ranged from 1.6 to 2.6 days for [224]Ra and from 2.4 to 5.6 days for [228]Ra. The tracer residence time can be estimated by dividing the radium inventory in surface waters by the sum of all losses (i.e., radioactive decay and exchange with offshore waters) (Rodellas et al., 2021). The assumption of steady state may therefore not be valid due to the significant difference between Ra activities from the first and second samplings (P1 and P2; Fig. 3), which were carried out only 4 days apart. Notice however that using a non-stationary Ra mass balance would have required monitoring the activities in coastal waters of Ra isotopes over the sampled period to understand its temporal patterns. Moreover, assuming steady state instead of a decrease of activities in coastal waters ($-da/dt$) (the pattern that was observed in the EPE from 2019; Fig. 3), results in conservative estimates of SGD induced by EPE relative to that in baseflow conditions.

### A.2.4 Endmember selection

Due to the large spatial variability of Ra isotopes in the groundwater activity at the experimental site of Argentona, constraining the Ra activity of the SGD endmember for both the brackish and saline components is particularly difficult. To overcome this limitation, endmembers were selected according to the following conditions: (1) the selection of the brackish Ra endmembers ($Ra_{BSGD}$) was constrained to groundwater samples with low salinities (Sal < 5), (2) saline endmembers were constrained to

groundwater samples with salinities higher than 5, and (3) the $^{224}$Ra/$^{228}$Ra activity ratios of both endmembers must satisfy the following equation:

$$\frac{^{224}Ra_{BSGD}}{^{228}Ra_{BSGD}} < \frac{F_{Ra224}}{F_{Ra228}} < \frac{^{224}Ra_{SSGD}}{^{228}Ra_{SSGD}}, \tag{A2}$$

where $F_{Ra-224}/F_{Ra-228}$ is the ratio between the total fluxes of $^{224}$Ra and $^{228}$Ra to the coastal ocean for each of the three

samplings (notice that if condition 3 is not met, the water flows resulting from the concurrent mass balance may be negative). A brackish and saline SGD was determined for each of any possible combinations between brackish and saline Ra endmembers that satisfied the above-mentioned conditions, and we reported the final SGD fluxes as the median value and the interquartile range. Conservative fluxes of nutrients were computed by multiplying the minimum concentrations of nutrients within the brackish and saline endmembers (discriminated by conditions 1 and 2), and the median ($\pm$ interquartile range) of each SGD

component (brackish and saline).

**B Darcy's law calculations**

The relative significance of EPE in annual SGD estimates derived from the Ra mass balance was compared with Darcy flux estimates ($Q = -k_h \cdot i$). For these calculations, hydraulic conductivity ($k_h$ [m s$^{-1}$]) was assumed to be in the order of $10^{-3}$ m s$^{-1}$ (characteristic of clean sands) in the range of local studies (personal communication T. Goyetche and L. Del Val). The

hydraulic gradient was determined as the difference between mean sea level (MSL) data and groundwater level data acquired from a CTD diver deployed in a piezometric well at the experimental site of the Argentona ephemeral stream (N3-15, 80 m from the shoreline). Absolute Darcy flux results (Fig. B1) should only be taken as indicative. In fact, the relative variation of Darcy flux during EPE can be used as a proxy for groundwater discharge. Results indicate that the EPE from October 2019, December 2019, and January 2020 represented 2, 4, and 6%, respectively, of the annual groundwater discharge. The relative

significance of EPE derived from these calculations is slightly lower than those obtained from the Ra mass balance. This discrepancy can be associated with the different discharge processes that each method captures. Whilst Ra mass balance enables the quantification of processes with different spatiotemporal scales and different compositions of groundwater (e.g., terrestrial groundwater discharge, porewater exchange), Darcy's law only captures the discharge of meteoric or brackish groundwater due to the hydraulic gradient at the shallowest aquifer.

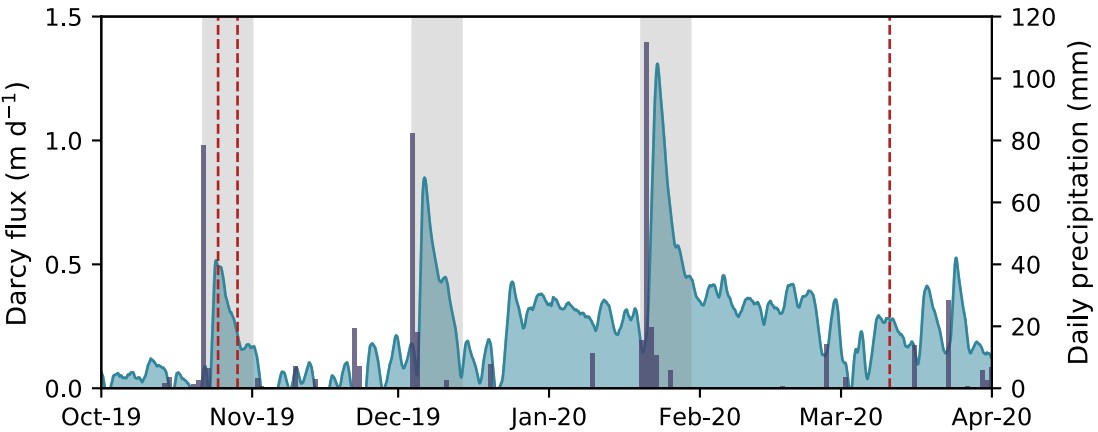

Figure B1. Temporal evolution of Darcy flux from October 2019 to April 2020 at the experimental site of the Argentona ephemeral stream. Red lines indicate the groundwater and seawater samplings performed at the study site (P1, P2, and BF) and grey bands indicate the major precipitation events of October 2019, December 2019, and January 2020.

## 7 Acknowledgements

This work was partly funded by the projects PID2019-110212RB- C22, CGL2016-77122-C2-1-R/2-R and PID2019-110311RB-C21 of the Spanish Government and the project TerraMar ACA210/18/00007 of the Catalan Water Agency. The authors want to express their thanks for the support of the Generalitat de Catalunya for MERS (2017 SGR-1588) and GHS (2017 SGR 1485) for additional funding. The authors would like to thank Maravillas Abad from ICM-CSIC for the analysis of nutrients. M. Diego-Feliu acknowledges the economic support from the FI-2017 fellowships of the Generalitat de Catalunya autonomous government (2017FI_B_00365). V. Rodellas acknowledges financial support from the Beatriu de Pinós postdoctoral program of the Generalitat de Catalunya autonomous government (2019-BP-00241). A. Alorda-Kleinglass acknowledges financial support from ICTA "Unit of Excellence" (MinECo, MDM2015-0552-17-1) and PhD fellowship, BES-2017-080740. Albert Folch is a Serra Hunter Fellow. We would like to thank all colleagues from the Grup de Recerca en Radioactivitat Ambiental de Barcelona - GRAB (Universitat Autònoma de Barcelona). We would like to thank SIMMAR (Serveis Integrals de Manteniment del Maresme) and the Consell Comarcal del Maresme for the construction of the research site.

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
