# Peer review of "Extreme precipitation events induce high fluxes of groundwater and associated nutrients to coastal ocean"

_Hydrology and Earth System Sciences, 2021_

## Author Response (AR1)

We sincerely thank reviewer 1 for his/her time in making constructive and helpful comments, which contributed to improve the overall quality of the manuscript. We have carefully addressed all the reviewer comments and have provided detailed answers to all queries. The comments made by the reviewer are quoted "in italics", our responses in black, and the modifications to the manuscript for the new submissions are quoted in **bold**.

On his/her general comments, the reviewer described his/her reservations about the SGD and nutrient flux calculations presented in this work, focusing on two categories: the definition of SGD source functions, and the mass balance of radium isotopes used for quantifying SGD. See general comments below:

*"The manuscript is well written and well-illustrated. Whilst the proposed implications of the results sound exciting and significant in the context of land-ocean exchange processes, I have serious reservations about the numbers put forward for nutrient fluxes. These reservations fall into two categories that are related:*

*1. A clearer, less ambiguous definition of different SGD fractions being considered ('terrestrial and marine' vs 'fresh and recirculated' must be put forward – one that is underpinned by the mechanics of flow through porous media.*

*2. A mass balance approach (steady state one at that) that discriminates between different flow components, as well as different source functions for nutrients must be justified more clearly considering the known issues with non-conservative behavior of both isotopic tracers and transported solutes, the non-linearity of the mixing process for radioisotopes, the number of degrees of freedom available for potential solutions for the source functions into the mass balance, and the nature of subterranean estuaries as biogeochemical hotspots.*

*I go into more detail on this with 2 queries that I would like addressed, but fundamentally, the approach followed appears (I might be mistaken, and in that case would be happy to be educated on the issue) to ignore a well know aspect of chemical reactor engineering, which applies if we think of the subterranean estuary as a chemical reactor mixing different inputs: This is that the mean age of the outflow mixture does not correspond to the average residence time of water masses within the coastal aquifer, especially if there is a change in the mixing regime, which is very likely given the impact of extreme precipitation events on subterranean estuary dynamics, purely from a mechanistic point of view. If we then take radioisotope ratios as fingerprints of distinct mixture components, and simultaneously as an indicator of mean water age allowing us to determine flushing time within the water volume receiving the SGD inputs, while assuming conservative behavior within the reactor (e.g., the subterranean estuary) for both isotopes and solutes being mixed, then the outcome of the budget trying to ascribe net nutrient transport into the coastal zone from both fresh and saline groundwater has to be uncertain."*

**Query 1: Definition of SGD source functions.**

In his/her first query, the reviewer emphasized the need for a "*clearer, less ambiguous definition of SGD fractions*", through a set of questions, which encompasses the terminology used (terrestrial and marine vs fresh and saline), the definitions of SGD fractions (based on salinity, Ra signature, and/or the underlying discharge mechanism), and the implications that this may have for SGD quantification (Ra mass balance). The reviewer points out a discussion, which we think relevant not only for the present study, but also for all research related with SGD, in view of the general lack of standardization of terminology and definitions regarding to this process. The following points stated by the reviewer are related to Query 1 and are clarified in the subsequent discussion (see reviewer extract below).

*"On Line 24: 'the flow of terrestrial and marine groundwater to the coastal ocean'.*

*The terrestrial and marine 'realms' are difficult to distinguish and define in a coastal aquifer. I would write fresh and saline (or salty) groundwater. However, it is unclear how this apparent distinction, made here, is reconciled with what is said in*

*Line 347: 'Whilst terrestrial SGD represents a net input of water to the ocean, marine SGD comprises disparate discharge processes solely involving the circulation of seawater through permeable sediments or the coastal aquifer', and*

*Line 358: "the study site, which are supplied via terrestrial (Combined discharge of meteoric groundwater and density-driven circulated seawater)".*

*Which is which: are the authors indicating that density driven circulation through the coastal aquifer is a net saline water input to the ocean? Are we separating inputs between fresh and saline, or are we distinguishing them based on Ra signatures, and therefore the need to include density driven circulation in the 'terrestrial' component?*

*But where is this density driven circulation happening? Is this in shallow sandy sediments, beach face, or is this the equivalent of return flow, and hence happening within the coastal aquifer at a larger spatial (but also temporal) scale?"*

Several studies in recent years have defined multiple mechanisms of groundwater discharge to the coastal ocean based on the composition of the groundwater flow, the underlying driving forces or/and its pathways (e.g., Garcia-Orellana et al., 2021; Santos et al., 2012; Taniguchi et al., 2019). According to these different features, SGD discharge processes are often classified in terrestrial groundwater discharge (the meteoric fresh groundwater flow originated from inland recharge and driven by terrestrial hydraulic gradients), density-driven seawater circulation (the flow of seawater associated with convection driven by thermohaline gradients originated due to the mixing of terrestrial and marine groundwater in the saltwater wedge of coastal aquifers), seasonal exchange of seawater (the flow of seawater driven by the movement of the freshwater-saltwater interface), shoreface circulation of seawater (the flow of recirculated seawater at the beach faces driven by tidal pumping or wave set up) and porewater exchange (the centimeter-scale exchange of groundwater through the water-sediment interface) (Garcia-Orellana et al., 2021; Michael et al., 2011; Robinson et al., 2018). The characteristics of the coastal alluvial aquifer of the Argentona ephemeral stream (see description of the study site in Section 2.1) favor the concurrent occurrence

of several SGD mechanisms, as shown in previous studies (Diego-Feliu et al., 2021; Folch et al., 2020; Martínez-Pérez et al., 2022; Palacios et al., 2019): meteoric seaward-flowing groundwater and recirculated seawater mix at multiple aquifer levels, which are separated by thin semi-confined layers of silt and clays (Folch et al., 2020; Martínez-Pérez et al., 2022), a seawater recirculation cell has been observed in the upper part of the aquifer (i.e., upper saline plume; Robinson et al., 2018), due to wave set up and/or sea level variations associated to recurrent extreme precipitation events (EPEs) (Palacios et al., 2019). EPEs induce movement of the freshwater-saltwater interface by promoting the offshore exchange of seawater (Palacios et al., 2019), and porewater exchange, which may also occur due to its almost ubiquitous character (Santos et al., 2012), although it has not been observed at the experimental site.

For convenience, we classify the above mentioned processes into 2 categories (notice that we cannot concurrently assess the discharge from all the individual mechanisms): Brackish SGD and Saline SGD. Brackish SGD includes the combined discharge of meteoric fresh groundwater and density-driven seawater circulating through the saltwater wedge (Terrestrial SGD in the original manuscript) and saline SGD includes the discharge of seawater circulating through permeable sediments (i.e. beach face recirculation, offshore exchange of seawater, and porewater exchange) (Marine SGD in the original manuscript). With the new definitions we avoid ambiguity of using the terms Terrestrial and Marine SGD which as stated by the reviewer '*are difficult to distinguish and define in a coastal aquifer*'. Moreover, these definitions avoid using the terms Fresh and Saline SGD, since fresh groundwater mixes with saline groundwater prior to discharge and therefore is not appropriate for this coastal aquifer. Notice that whilst the brackish SGD in this study can be predominantly referred as a long-scale SGD pathway, the saline component comprises disparate mechanisms but predominantly governed by short-scale flow processes (minutes to days). Separating SGD fractions based on spatiotemporal scales is crucial for both applying Ra isotopes as tracers and estimating meaningful SGD-driven nutrient fluxes.

The reviewer also expressed its reservation on the "*definitions presented throughout, as they translate into the mass balance approach (A1) and might affect the suitability of conclusions*" (see extract below).

"*These questions stem from the same issue: the definition of 'terrestrial' and 'marine' SGD is ambiguous. They are important, because any answer has consequences in terms of the way nutrient inputs to the coastal ocean are estimated and more importantly whether those estimates are valid: while fresh groundwater is a net input of water into the ocean, saline inputs are the result of a circulation cell of some type, so over the period of the circulation process, there is no net water input?*

*From Line 173: 'Here, we define terrestrial groundwater discharge as the combined discharge of meteoric groundwater and density-driven circulated seawater, and marine groundwater discharge as those processes solely involving the circulation of seawater through permeable sediments (i.e., beach-face circulation, porewater exchange).'*

*I have reservations on the clarity of definitions presented throughout, as they translate into the mass balance approach (A1) and might affect the suitability of conclusions. It is clear to me that the authors are separating the components based on Ra signatures, more specifically the 224/228*

*ratios. How they then reconcile this separation made based on an isotope signature with the mechanics of water flow through the coastal aquifer, which defines origins, pathways and whether a net water input into the ocean exists impacts on the credibility of the conclusions."*

We believe that the definitions of the two main SGD components presented above are clearer and less ambiguous relative to the ones presented in the original manuscript. Notice however that the quantification technique used in this study (Ra isotopes) also shaped the way we defined the different SGD fractions. Since any technique for quantifying SGD is targeting a specific discharge process or a set of them (e.g., water balances target fresh SGD), Ra isotopes are not an exception of that. Actually, the fact that we divided the components in brackish and saline SGD is partially because the pair of Ra isotopes used, $^{224}$Ra ($T_{1/2}$=3.66 d) and $^{228}$Ra ($T_{1/2}$=5.75 y), are commonly applied together for tracing short and long scale SGD (e.g., Alorda-Kleinglass et al., 2019; Rodellas et al., 2017; Tamborski et al., 2017). This is possible because groundwater is enriched in these isotopes in a rate which depends on their specific half-life. Therefore, whilst groundwater is generally enriched in $^{224}$Ra after very small- (centimeters to meters) and short-scale pathways (seconds to minutes), long groundwater flow paths and transit times are needed for groundwater to be significantly enriched in $^{228}$Ra (Garcia-Orellana et al., 2021; Michael et al., 2011). Consequently, $^{224}$Ra is targeting any SGD process and $^{228}$Ra only those with high spatiotemporal scales (i.e., $^{228}$Ra is not properly targeting porewater exchange or beach face circulation of seawater). We therefore believe that the use of Ra isotopes, in the way we use them in this study, enables reporting meaningful water flows, especially when using these water flows for quantifying nutrient fluxes. Notice that is convenient to separate SGD fractions based on temporal scales since the nutrient transformations occurring within the subterranean estuary are highly dependent on the temporal and spatial extent of the flow paths. Notice also that the methodology applied here represents a step further relative to the vast majority of SGD studies, which generally report total SGD fluxes (some of them based on short-lived Ra isotopes mass balances), which are commonly useless when converting them to a total flux of nutrients.

In the new version of the manuscript we will introduce some of the information regarding SGD pathways and definitions of fractions to clarify the points raised by Reviewer 1. The description of pathways at the study is mainly based on the information presented in Diego-Feliu et al. (2021), Folch et al. (2020), Martínez-Pérez et al. (2022), and Palacios et al. (2019) and will be included in section 4.1.1 as follows:

**"4.1.1 Pathways of submarine groundwater discharge**

**Submarine Groundwater Discharge incorporates a set of water flow processes involving the discharge of fresh groundwater and the circulation of seawater through permeable sediments (Garcia-Orellana et al., 2021; Michael et al., 2011; Santos et al., 2012). The driving forces and pathways of these processes likely determine the extent of the chemical reactions occurring in the subterranean estuary (Moore, 1999). Therefore, considering all the different SGD pathways concurrently occurring in a specific study site is fundamental for deriving reliable estimates of SGD and associated nutrient fluxes (Garcia-Orellana et al., 2021). The characteristics of coastal alluvial aquifers linked to the presence of ephemeral streams in the Maresme county may favor the concurrent occurrence of different water flow processes. Indeed, previous works conducted at the study site of Argentona have already shown that different SGD components coexist (Diego-**

Feliu et al., 2021; Folch et al., 2020; Martínez-Pérez et al., 2022; Palacios et al., 2019). Meteoric groundwater flowing seaward and recirculated seawater mix in multiple aquifer levels, which are separated by semi-confining thin layers of silt and clays. The different aquifer units and mixing zones may promote the combined discharge of fresh and saline groundwater (brackish) at the coastline or far beyond depending on the continuity of the confining layers (Folch et al., 2020; Martínez-Pérez et al., 2022). A seawater recirculation cell has been observed in the upper part of the aquifer (i.e., upper saline plume; Robinson et al., 2018), where seawater infiltrates through the shallower layers due to wave set up and/or sea level variations associated to extreme precipitation events (EPEs) or storm surges (Palacios et al., 2019). Offshore seawater exchange due to the movement of the freshwater-saltwater interface may also occur in response to the increased infiltration of rainwater inland associated with EPEs. Interflow may occur after an EPE, however it may probably reach easily the water table due to the thin vadose zone (2 to 3 m) and the high permeability of the surficial materials (Martínez-Pérez et al., 2022). Porewater exchange may also occur due to its almost ubiquitous character and the disparate mechanisms driving the water flow (Santos et al., 2012).

In this work, for convenience, the water flow processes described above have been clustered into two main SGD components: brackish and saline SGD. Brackish SGD is defined here as the combined discharge of meteoric groundwater and (long-term) density-driven recirculation of seawater through the saltwater wedge, regardless of the mixing degree between the two water masses and the aquifer unit considered. It should be noticed that this SGD component (1) does not represent a net water input to the coastal ocean, since it comprises a fraction of recirculated seawater (2) exclude water flow processes solely involving the short scale recirculation of seawater through permeable sediments, and (3) also include the contribution that interflow may have on groundwater discharge after the occurrence of an EPE. On the other hand, beach-face recirculation of seawater through the upper saline plume, porewater exchange, and offshore exchange of seawater due to the movement of the saltwater wedge are ascribed here to the Saline SGD component. This SGD component (1) represents a net zero water input to the coastal ocean for timescales longer than that of the process driving its oscillations, and (2) comprises a set of water flow processes with disparate spatiotemporal scales between minutes to days."

**Query 2**

In its second query, the reviewer expressed its reservations regarding the radium mass balance approach for quantifying water flows and associated nutrient fluxes for the two SGD components described above. The reservations lie in different categories including: how to ascribe and differentiate the SGD pathways? Are these pathways/sources actually distinguishable? Are the assumptions taken in the water and nutrient fluxes quantifications justifiable (net nutrient fluxes, end-member selection, steady-state, etc)? In the following discussion, we seek to clarify all the issues regarding the model conceptualization and the uncertainties of the radium mass balance used in this work for quantifying SGD and nutrient fluxes.

SGD components discrimination

Reviewer extract:

*"The authors try to ascribe different isotopic signatures to both components, as per Line 187: 'Both trends may indicate that the relative contribution of the terrestrial component of SGD, which is characterized by 224Ra/228Ra ARs close to the equilibrium value (1.0 to 2.2; Diego-Feliu et al., 2021), increased during the occurrence of the EPE.' End-member selection is explained in section A.2.4., but are the two components separable? If they are, then it should be clearly explained how this was done, since there are multiple solutions explaining the measured isotopic ratios in the coastal volume that was sampled."*

Concurrent mass balances for $^{224}$Ra and $^{228}$Ra are used in this study for distinguishing brackish and saline SGD components as defined above, "*but are these two SGD components separable*", as it is done in this study? The question, raised by the reviewer (see extract above), is addressed here through the following considerations:

1. It is well-known that different spatiotemporal components of SGD have specific Ra isotopes signature based on the half-lives of these radionuclides (Garcia-Orellana et al., 2021; Taniguchi et al., 2019).
2. Independent single-radionuclides mass balances for $^{224}$Ra and $^{228}$Ra performed at the study site during the same campaigns (calculations not included in the manuscript) resulted in higher water flows when using $^{224}$Ra relative to that resulting from $^{228}$Ra mass balance. This emphasizes that $^{224}$Ra captures a wider range of SGD processes than $^{228}$Ra, which may only capture long spatiotemporal scale SGD pathways (Garcia-Orellana et al., 2021).
3. Although, as stated by the reviewer "*there are multiple solutions explaining the measured isotopic ratios in the coastal volume that was sampled*", we believe that the significant increase in $^{228}$Ra (predominantly delivered by long scale SGD processes) relative to $^{224}$Ra after the extreme precipitation event occurred in October 2019 is indicative of higher brackish (long-scale) SGD flow rates (with higher $^{228}$Ra to $^{224}$Ra activities relative to short scale recirculated seawater processes).
4. The criteria for "*ascribing different isotopic signatures to both components*" has been the use of a quasi-arbitrary value for salinity in the groundwater endmembers. Whilst we attributed groundwater salinities below 5 to the brackish component of SGD, groundwater endmembers presenting higher salinities were attributed to saline SGD. This criterion/assumption is also based on the observed trends in groundwater salinity at the study site, whereby piezometric wells located at the freshwater-saltwater interface or below are characterized by low salinities (<5).

Despite the above-mentioned considerations (1 to 4), we are aware of the discretional character of separating the SGD components in the way we did in this study. We are also aware of the possible biases and conceptual uncertainties that may derive from the arbitrary decisions taken in the quantification process. However, we do believe that the procedure chosen in this study is (1) the more accurate way for quantifying meaningful water flows by means of Ra isotopes at the study site, and (2) it enables assessing the relative significance of EPEs, besides the possible uncertainties in absolute water flows quantifications, which is the main goal of the present manuscript. In the new version of the manuscript, we will emphasize possible biases and uncertainties (see extract below, Section 4.1.2), and we will indicate more precisely the criteria for endmember selection. However, we will be very pleased to discuss and integrate in future versions of the manuscript

alternative conceptual models, methods for quantifying, or uncertainty assessments based on reviewer's recommendations.

In the new version of the manuscript, we will include the following text:

**"In baseflow conditions, the brackish component of SGD (including fresh groundwater and density-driven seawater discharge) represented 60% of the total SGD (Fig. 4). The relative contribution of this SGD component increased after the rainfall event of October 2019 to up to 75% of the total SGD. This is consistent with the variation on the $^{224}$Ra/$^{228}$Ra AR in coastal seawater after the EPE (see Section 4.1.1) and coherent with Darcy's flow calculations (Appendix B). These estimates of the relative contribution of the brackish component are generally much larger than estimates of fresh groundwater discharge for the Mediterranean Sea (1 - 25%, Rodellas et al., 2015), global estimates (10%, Kwon et al., 2014; 0.06%, Luijendijk et al., 2020), and local studies (5 - 55%; Alorda-Kleinglass et al., 2019; Beck et al., 2008; Kiro et al., 2014; Knee et al., 2016; Rodellas et al., 2017; Tamborski et al., 2017). This difference most likely emphasizes that whereas the studies presented above are mainly focused on distinguishing fresh and saline SGD, here we are targeting brackish (encompassing meteoric groundwater and recirculated seawater) and saline SGD, as previously discussed. It should also be noticed that the estimates presented in this study should be taken as semi-quantitative in view of the biases, limitations, and uncertainties discussed in detail in appendix A (e.g., endmember selection, steady-state assumption, lack of consideration for runoff). However, these limitations are inherent to almost any SGD study and especially those using Ra isotopes as tracers (Garcia-Orellana et al., 2021; Rodellas et al., 2021), and may not invalidate the implications derived from this study, which is to determine the relative significance of EPEs in water and nutrient fluxes to the coastal ocean."**

Net water input to the coastal ocean

The approach used in this study does not allow the discrimination between fresh (or net groundwater inputs) and saline SGD. As stated by the reviewer "*for the calculation of nutrient flows into the sea, the net input of water matters the most, and importantly, the fact that the composition of the flows is changed by the mixing, the circulation path, and the timing of the process.*". We do not believe that the net water input matters the most regarding the transport of nutrients from the coastal aquifer to the coastal ocean. In fact, nutrient inputs associated with saline SGD are estimated to be orders of magnitude higher than that relative to fresh groundwater inputs (Santos et al., 2021). Instead of determining the net water flow, which may be important for hydrological balances, but clearly uncertain when determining nutrient fluxes (see comments query 1), we resolved to quantify the combined discharge of meteoric groundwater and long-scale recirculated seawater. However, nutrient fluxes should be taken as a first-order approximation since many assumptions (discussed below) were taken to translate the water flow to the nutrient flux.

Net nutrient inputs to the coastal ocean

In one of its remarks, the reviewer indicated that net inputs of nutrients are not determined for both SGD components considered:

*"We have two components of that mixture: one that is circulating through the coastal aquifer and is therefore characterized by a spectrum of groundwater residence times and biogeochemical histories, and the other that is dragged along and/or forced by the hydraulic gradient and is fresher, but is also characterized by a distribution of residence times. To extract a net flux of nutrients into the ocean arising from the first process, one would need to determine the difference between the concentration at the beginning of the loop (what goes into the coastal aquifer from the sea) and the one at the end of the loop (what comes out after residing in the coastal aquifer), as well as the discharge corresponding to the circulation flow. This is not done."*

and:

*"For the second process, one would then have to determine the discharge associated with the net amount of water (fresh) incoming to the ocean, as well as the concentration of nutrients within that water mass."*

We agree with the reviewer's observation and in the new version of the manuscript we will indicate that the reported nutrient fluxes are not net nutrient fluxes (see extract below, Section 4.1.3). However, it should be noticed that since the concentrations in the endmembers are orders of magnitude higher than that of the sea ("the concentration at the beginning of the loop (what goes into the coastal aquifer from the sea)"), the relative significance of determining the net flux of nutrients rather than our calculations is almost negligible. For instance, median seawater concentrations of DIN, DSi, and DIP only represent 0.3, 1, and 4% of the minimum concentration of the brackish SGD endmembers, respectively. This contribution is higher for the saline SGD endmembers (13, 2, and 4%, respectively), yet by using as endmembers in both cases the minimum nutrient concentration in groundwater, we are reporting conservative nutrient fluxes, and extracting the inputs from the sea may not significantly change the magnitude of these fluxes.

**"The SGD-driven nutrient fluxes were estimated by considering the brackish and saline Ra-derived SGD flows and the respective nutrient concentration in groundwater from both fractions (see Appendix A). Notice that the obtained results are not expressed here in terms of net nutrient inputs since the fluxes of nutrients from the coastal ocean to the coastal aquifer are not considered. However, the influence that these fluxes have for the calculations may be negligible since concentrations of nutrients in seawater are orders of magnitude lower than those in groundwater (see SI; Fig. S2)."**

The reviewer also expressed its reservations regarding nutrient fluxes calculations in several of its comments:

*"Even so, this would ignore the fact that the two components mix, and hence the chemical makeup of the solution that comes out cannot easily be reconstructed, and certainly not by assuming linear bi-component mixing. Regardless, we are also assuming here that the nutrients themselves are conservative through all the process and hence the two water masses can be distinguished not only by their isotopic composition, but also by their nutrient composition. It is not clear to me how this is done."*

and,

*"So, fluxes cannot be calculated tout-court by multiplying an apparent water mass flux (FFSGD and FRSGD above, however they are calculated) by the 'end-member' nutrient concentration.  This approach not only assumes that the transported radioisotopes are a) conservative, b) mix linearly across the domain and this can therefore be treated like a bi-component mixture, but also that c) nutrients are conservative, and d) it is possible to ascribe a unique source composition to each endmember, which is difficult because a) and b) are not verified."*

We are aware of the limitations that the study presented here have in terms of accurately reporting nutrient fluxes associated with SGD. We also recognize that these limitations are inherent to almost any SGD study, as discussed in many published articles (e.g., Cerdà-Domènech et al., 2017; Cho and Kim, 2016; Rodellas et al., 2021; Santos et al., 2021). In that sense, we would like to emphasize that the reported nutrient fluxes can be considered as semi-quantitative in view of the obvious uncertainties and limitations of the nutrient fluxes calculations, and we will make this evident in the next version of the manuscript (see extract below, Section 4.1.3). However, we believe that these uncertainties and limitations do not invalidate the implications derived from this study because: (1) the reported nutrient fluxes are likely conservative since we have used the minimum groundwater nutrient concentration for quantifying these fluxes, and (2) the aim of this study is not the accurate assessment of nutrient fluxes to the coastal ocean, which would require further study of all possible nutrient transformations within the subterranean estuary, but the assessment of the relative significance of water and nutrients supply to the coastal ocean during EPEs.

It should be noticed that nutrient fluxes were estimated by multiplying the volumetric water flow of brackish and saline SGD by the minimum nutrient concentration from a set of onshore samples, selected following the criterion used for the Ra endmembers, as explained in the appendices (see appendix A.2.4). Since it was not possible to directly collect the discharging groundwater, by using onshore samples we are implicitly assuming that no nutrient transformation occurred between the sampling and discharging points, along the subterranean estuary (Cook et al., 2018). This assumption is perhaps one of the main sources of uncertainty in the reported nutrient fluxes as it has already shown by many other authors (Sawyer et al., 2014; Weinstein et al., 2011; Wong et al., 2020). It should also be noted that these SGD-derived nutrient estimates may be biased due to the groundwater endmember selection, since nutrient concentrations in discharging groundwaters may vary during EPE due to dilution, increasing lixiviation of fertilizers, or enhancement of biogeochemical reactions in the mixing zone of coastal aquifers (Spiteri et al., 2008). Although all the assumptions made for nutrient fluxes quantification may result in high degrees of uncertainty, the results presented in this study enable the assessment of EPE significance as a major driving force transporting nutrients to the coastal ocean."

**Technical edits:**

Here we discuss the technical edits made by the reviewer 1:
Reviewer comment:

*"Line 17: 'Results indicate that the groundwater flows of terrestrial and marine SGD after the extreme precipitation event were 1 order of magnitude higher than those in baseflow conditions.'*

*I fail to see a mechanism explaining here how the saltwater (marine) SGD flows increased driven by an EPE. The classifications of 'terrestrial' and 'marine' are ambiguous in the context of SGD and should be clearly grounded on the mechanics of groundwater flow. See also specific queries."*

Answer: Higher Saline SGD may result from different mechanisms: (1) increasing seawater circulation on the beach face due to higher wave heights, (2) increasing porewater exchange due to increased wave pumping, and (3) increasing exchange due to movement of the freshwater-saltwater interface seawards.

Reviewer comment:

*"Line 50: 'Infiltrated water displaces groundwater stored in the aquifer towards the sea, enhancing mixing processes in the coastal aquifer'*

*This is not entirely correct. The fact that precipitation percolates through soil does not guarantee it reaches the local water table, thus adding its mass to the freshwater body in the aquifer; this is when the second part of the sentence would apply. The role of interflow is not well understood, and the timing of flow through the unsaturated zone varies tremendously (well beyond the scale of EPEs anyway), depending on geology, soil type, land cover, surface gradient, accumulated precipitation, and degree of clogging as well as precipitation rate – so this sentence must be rewritten. What fraction of 'terrestrial' SGD is interflow?"*

Answer: We thank the reviewer's suggestion, and we will rewrite the sentence according to the current knowledge about the effects of EPE in coastal aquifers. It is certain that the infiltration of rainfall may not result directly in enhanced mixing processes, so we will rephrase the sentence excluding the second part (see extract 1 below, Section 1). Also, we will introduce more detail regarding the description of the coastal aquifer of the Argentona ephemeral stream since it may be important to understand its response to EPEs (see extract 2 below, Section 2.1). The vadose zone of the aquifer in its lower part (where the experimental site is located) is only about three meters depth and the materials are very permeable (Martínez-Pérez et al., 2022). This may promote infiltrated rainfall to reach the water table as it can be observed in Figure 2, where groundwater level rises 60 cm as a response to the EPE of October 2019. The role of interflow, as stated by the reviewer is not understood, and there is no way we could distinguish which fraction of the brackish SGD is associated with this flow. However, based on groundwater levels of different piezometric wells (data not shown in the manuscript) and also based on data from cross-hole electric resistivity tomography performed by Palacios et al. (2019) at the study site reveals that EPEs increase groundwater level and promotes the movement of the freshwater-saltwater interface towards the sea. This is likely to induce the discharge of 'old water' contained in the aquifer, whether brackish or saline. In the new version of the manuscript we will make clear that it has not been possible to distinguish the fraction of brackish SGD associated with interflow relative to that of 'old water' (see extract 3, Section 4.1.1).

**Extract 1: "Extreme precipitation events may indeed promote aquifer recharge through the infiltration of rainwater (Ramos et al., 2020; Yu et al., 2017), although its effects on piezometric levels (quantitively and temporally) depend on several factors, such as soil composition, geological characteristics, the hydraulic parameters of the aquifer, and others. Infiltrated water**

displaces groundwater stored in the aquifer towards the sea, and in some cases may also enhance mixing processes in the coastal aquifer (Anwar et al., 2014; Palacios et al., 2019; Robinson et al., 2018)."

Extract 2: "The phreatic level in the lower part of the Argentona ephemeral streams is shallow (2 to 3 meters below the ground level) and the materials are highly permeable (Martínez-Pérez et al., 2022). This facilitates the rapid aquifer recharge after an EPE, since infiltrated rainwater may easily reach the water table, and diminishes the role of interflow circulating through the vadose zone."

Extract 3: "Brackish SGD is defined here as the combined discharge of meteoric groundwater and density-driven (long-term) recirculation of seawater through the saltwater wedge regardless of the degree of mixing between the two water masses and the aquifer unit considered. It should be noticed that this SGD component (1) does not represent a net water input to the coastal ocean, (2) exclude water flow processes solely involving the recirculation of seawater through permeable sediments, and (3) also include the contribution that interflow may have on groundwater discharge after the occurrence of an EPE."

Methods

"Line 106: 'as well as seawater samples' – clarification needed. Temperature and salinity measured in samples taken at sea as well? The sentence is not clear."

We will modify the sentence according reviewer's suggestion, in the new version it reads:

"Salinity and temperature of groundwater and seawater samples were measured in-situ with two handheld probes (HANNA HI98192 and WTW COND 330I)."

"Line 125: 'Polyethylene vials' – clarify. HDPE is the standard for nutrient analysis. Was this used, or simple polyethylene vials? "

The samples were collected in HDPE vials, this has been indicated in the manuscript.


[revised manuscript text omitted]

**Anonymous referee #2**

We sincerely thank anonymous reviewer 2 for his/her helpful and constructive comments. In his/her remarks, the reviewer expressed the need for clarifying the assumptions and limitations related with the quantification of SGD and nutrient fluxes through the Ra mass balance. Specially, the reviewer focused in the steady-state assumption and on the lack of consideration of runoff as a source of Ra isotopes and nutrients to the coastal ocean. See his/her general comment below:

*"The authors used a radium survey to assess inputs of submarine groundwater discharge (SGD) to the Mediterranean Sea in northeastern Spain following an extreme precipitation event and at base flow. They showed that terrestrial water inputs increased by an order of magnitude 4 days after the storm and returned to base flow conditions another 4 days later. The episodic terrestrial and marine nutrient inputs associated with this one event likely accounted for more than 10% of the dissolved inorganic nitrogen, dissolved inorganic phosphorus, and dissolved silicate inputs to the coast for the whole year. This highlights the importance of extreme events for nutrient inputs to the coast.*
*The study will be of great interest to those who study nutrients in coastal waters. It is one of a relatively small number of studies to quantify changes in submarine groundwater discharge and associated nutrients during large recharge/rainfall events. I have minor suggestions to improve the clarity of the manuscript. The most important is the need for the authors to more clearly communicate the assumptions in their Ra and nutrient budget analyses within the methods and discussion, rather than the appendix. In my view, the most severe limitations are the steady-state assumption and the lack of consideration of runoff as an input.*

*Regarding runoff, flow occurred in at least some of the ephemeral streams at T1, T2, and T3 during the October 2019 event (L 372). This is important information and should be stated in the main text rather than the appendix. Was runoff water sampled for Ra isotopes and nutrients? The authors argue they can neglect runoff in their Ra and nutrient budgets because the Ra delivered by overland flow may have decreased by 90% at the time of the P1 sampling, but if the total delivery was large, 10% of that total delivery could still be sizeable. Ideally the authors would perform some calculations to examine the potential scale of the runoff contribution. Did the authors collect any runoff samples for Ra isotope and nutrient analysis? The volumetric flux of runoff is likely unknown, but an estimate could probably be made based on typical runoff ratios for the region and the known catchment area. Without this kind of a calculation, the assumptions and limitations of lacking these runoff measurements should be clearly discussed. Care should be taken in attributing all terrestrial water inputs to groundwater (as in L 210) and all terrestrial nutrient inputs to groundwater (as in L 228). A large amount of sediment-water interactions would be expected in a flowing, turbid ephemeral stream under an extreme precipitation event, so the contribution of runoff to radium isotopes and nutrients should not be readily discounted without further analysis."*

According to the reviewer's suggestions, in the new version of the manuscript we will address in detail the runoff produced by the EPE and we will introduce the following information in the text:

1.  The description of the EPE occurred in October 2019, including the runoff associated with this episode, will be introduced in the main text of the manuscript rather than in the appendix.

2. An estimate of runoff during the EPE based on a soil mass balance will be included in the new version of the manuscript.
3. An estimate of the Ra flux associated to episode will also be included in the new version of the manuscript
4. The influence that runoff may have had on the calculations of SGD and nutrient fluxes will be discussed in the main text and in the appendix of the new version of the manuscript.

Description of the flash flood events in the Argentona ephemeral stream

The hydrological regime of Mediterranean ephemeral streams have been described in detail in many research articles due to the hazardous characteristics of its associated flash flood events (e.g., Ballesteros et al., 2018; Camarasa-Belmonte and Tilford, 2002; Colombo and Rivero, 2017). Surface runoff in the Argentona ephemeral stream only occurs after heavy rainfall events characterized by its short duration and great intensity. The conceptual hydrograms corresponding to these events are well known and have been described in detail especially in grey literature (Cisteró and Camarós, 2014; Riba, 1997). According to these hydrograms, the flood events consist of different stages, which take place in few hours (2 to 6) depending on the intensity and duration of the EPE: (1) some minutes after the precipitation has started a thin layer (some cm) of 'dirty' water (from the surroundings) flow superficially towards the sea, (2) after some minutes to hours, a cleaner water mass, which carries heavier materials overcomes the first one, (3) the flood level increases progressively towards a maximum discharge rate, which remains constant for a short period of time, and (4) the water level decreases gradually until completely disappearing (Cisteró and Camarós, 2014). Water velocities in one of the heaviest precipitation events occurred in the Argentona ephemeral stream (180 mm in 24 h) was calculated to be on the order of 2.7 to 3.8 m s$^{-1}$ (Martín-Vide, 1985). This information is now presented in the description of the study site of the new version of the manuscript (Section 2.1):

**"In this region, most of the ephemeral streams are hydraulically disconnected from their alluvial aquifers and, therefore, surface runoff takes place only after the most significant rain events, which are characterized by short duration and high rainfall intensity. The nature of floods associated to EPEs are well known and have been described in detail especially in grey literature (Cisteró and Camarós, 2014; Riba, 1997). Floods associated to the EPE events consist of different stages, which take place within a few hours (2 to 6 h) depending on the intensity and duration of the EPE: (1) a thin layer (a few centimeters) of "dirty" water (from the surrounding area) flows towards the sea a few minutes after the onset of rainfall, (2) then, after some minutes to hours, a cleaner water mass carrying heavier materials, flows along the ephemeral stream; (3) the flood level increases progressively towards a maximum discharge rate, which remains constant for a short period of time (some hours), and (4) the water level decreases gradually until completely disappearing (Cisteró and Camarós, 2014). Water velocities in one of the heaviest precipitation events occurred in the Argentona ephemeral stream (180 mm in 24 h) was calculated to be on the order of 2.7 to 3.8 m s$^{-1}$ (Martín-Vide, 1985)."**

Surface runoff in the Argentona ephemeral stream on October 22$^{nd}$, 2019

Surface runoff has been estimated by soil mass balance (based on type of soil, land use, geology, precipitation, slope, etc.) during the rainfall event of October 22$^{nd}$, 2019. The soil mass balance has

been used for a calibrated regional groundwater numerical model of the southern section of Maresme county. The model is not publicly available, as it has been developed for a specific work of the Spanish railway public company (ADIF). According to the soil mass balance, surface runoff associated to this EPE was about 1 hm$^3$. This information is now mentioned in Section 3.1 Meteorological and hydrological context:

**"Estimating the runoff velocity and discharge in the study site is difficult because in the midd-19$^{th}$ century a set of galleries and dams were constructed at the upper part of the Argentona ephemeral stream (municipality of Dosrius) to collect groundwater and surficial water from this area. The effect that these structures may have on surface runoff is uncertain. However, a soil mass balance (based on type of soil, land use, geology, precipitation, slope, etc.) of the lower part of the Argentona ephemeral stream has been used to provide a semi-quantitative estimate of surface runoff during the October 22$^{nd}$, 2019 rainfall event. The soil mass balance has been used for a calibrated regional groundwater numerical model of the southern section of Maresme county. The model is not publicly available, as it has been developed for a specific work of the Spanish railway public company. According to the soil mass balance, surface runoff associated to this EPE was about 1 hm$^3$. "**

Radium inputs due to surface runoff

A proper monitoring of Ra isotopes activities and nutrient concentrations in runoff water during the EPE occurred in October 2019 was not done. This is due to the difficult sampling situation and the dangers inherent to flood events. However, a single surface water sample was taken at the initial stage of the flood, when the amount of water flowing represented only a thin layer of water (some cm). A first-order assessment of the Ra flux associated with surface runoff in the study site was performed by using the $^{224}$Ra and $^{228}$Ra activities of this water sample and the calculated runoff discharge. To assess the influence of this Ra input on our estimates of groundwater discharge, we have used a non-stationary Ra mass balance, which considers (1) a punctual surface input of Ra, (2) the offshore exchange of Ra, and (3) radium decay:

$$A(t) = A_{runoff}\, e^{-t\left(\frac{1}{T_F}+\lambda\right)}$$

Eq. 1

where A [Bq] is the radium activity in seawater at a specific time (t [d]), A$_{runoff}$ [Bq] is the activity delivered by the punctual input through surface runoff and it is calculated by multiplying the total surface runoff (Q [m$^3$]) by the specific activity of Ra in surface water (a [Bq m$^{-3}$]), T$_F$ is the flushing time of radium, and $\lambda$ is the decay constant of each radionuclide. Notice that for $^{228}$Ra, mixing with offshore water may be the predominant output and decay losses can be neglected. Equation 1 has been used to determine the activity of both Ra isotopes in seawater in the two subsequent samplings performed after the EPE and compared with the seawater Ra inventories found during these two samplings.

The relative significance of runoff derived from the EPE in the Ra inventories based on the above calculations is 2 and 1% for $^{224}$Ra, and 1 and 8% for $^{228}$Ra at the first and second sampling, respectively. Notice that these values are low and comparable with the common uncertainties derived from the measurement of Ra isotopes. It should be also noticed that the calculated Ra

inputs through surface runoff are likely overestimated since the surface water sample was collected at the beginning of the flood and it is representative only of the initial thin and 'dirty' water (with more particles per mass of water) flow and not of the 'cleaner' water mass, which represents most of the total runoff discharge. Therefore, we do believe that this punctual source of Ra is negligible and may not affect the estimations regarding SGD and nutrient fluxes made in this article. In the new version of the manuscript we will mention that the estimates may be slightly biased towards higher SGD and nutrient fluxes due to the lack of consideration of surface radium and nutrient inputs (see extract Section 4.1.2).

**"It should also be noticed that estimates presented in this study should be taking as semi-quantitative in view of the biases, limitations, and uncertainties discussed in detail in appendix A (e.g., endmember selection, steady-state assumption, lack of consideration for runoff)."**

We also included the discussion about the role of runoff in the quantification of SGD and nutrient inputs in the appendices:

**"Ra inputs from runoff water were also discarded for the sampling conducted in March 2020 (BF) due to the total absence of surface water inputs during the sampling period. In October 2019, 4 days before the first sampling conducted at the study site, runoff occurred in direct response to an EPE (~90 mm). However, considering the flushing time of Ra isotopes in the coastal system (see S1.2.2), the Ra delivered by this punctual runoff may have decreased by >90% for the first sampling (P1) and by >99% for the second sampling (P2), due to decay (for $^{224}$Ra) and mixing with offshore waters. However, the relative contribution that runoff may have had on Ra inventories during the first two samplings was calculated using the Ra activities of the runoff sample collected during the beginning of the EPE, and the calculated runoff discharge by using a soil mass balance. Although the estimates may be uncertain, the results indicate that the relative significance of runoff derived from the EPE in the Ra inventories were 2 and 1% for $^{224}$Ra, and 1 and 8% for $^{228}$Ra at the first and second sampling, respectively. Notice that these values are low and comparable with the common uncertainties derived from the measurement of Ra isotopes. It should be also noticed that the calculated Ra inputs through surface runoff are likely overestimated, since the surface water sample was collected at the beginning of the flood and it is representative only of the initial thin and 'dirty' water (with more particles per mass of water) flow and not of the 'cleaner' water mass, which represents most of the total runoff discharge. Therefore, Ra runoff input was discarded as a major source of Ra isotopes to the coastal ocean during the sampling periods."**

Regarding stationary or transitory conditions of the Ra mass balance, we consider that (1) using a non-stationary Ra mass balance would have required monitoring the activities of Ra isotopes over the sampled period, a sampling effort that was not possible to conduct, and (2) the assumption of steady state may result in conservative estimates of SGD induced by EPEs relative to that in baseflow conditions. We have indicated the assumption taken for solving the model throughout the manuscript (Section 4.1.2 and 4.2.3) and in appendices (see extract below):

**"A.2.3 Steady state conditions**
**Steady state conditions (i.e., tracer inventories do not vary with time; $da/(dt \cdot V)$ = 0) are often assumed in Ra mass balances (e.g., Alorda-Kleinglass et al., 2019; Beck et al., 2008; Rodellas et al.,**

**2017). This assumption implies that Ra inputs and outputs are balanced for a time period equivalent to the tracer residence time in the system (Rodellas et al., 2021). In Maresme County, the tracer residence time ranged from 1.6 to 2.6 days for $^{224}$Ra and from 2.4 to 5.6 days for $^{228}$Ra. The tracer residence time can be estimated by dividing the radium inventory in surface waters by the sum of all losses (i.e., radioactive decay and exchange with offshore waters) (Rodellas et al., 2021). The assumption of steady state may therefore not be valid due to the significant difference between Ra activities from the first and second samplings (P1 and P2; Fig. 3), which were carried out only 4 days apart. Notice however that using a non-stationary Ra mass balance would have required monitoring the activities in coastal waters of Ra isotopes over the sampled period to understand its temporal patterns. Moreover, assuming steady state instead of a decrease of activities in coastal waters ($-da/dt$) (the pattern that was observed in the EPE from 2019; Fig. 3), results in conservative estimates of SGD induced by EPE relative to that in baseflow conditions."**

**Technical edits:**
Here we discuss the technical edits of reviewer 2:

*"The spatial relationships in the study could be clarified in a couple of places. For example, L 99 and Figure 1 refer to the Medistraes project or site. Is Medistraes an alternate name for the Argentona site? If so, it would be clearer in the figure and text to just refer to the site location by one name (or else label Medistraes project on Figure 1b). L 157 refers to "groundwater from the site of the Argentona ephemeral stream." I would suggest calling this the "Argentona site" and refering to Figure 1c-d for clarity."*

We agree with the reviewer on the need for standardization of the terms used to describe the study site. From now on, in the new version of the manuscript we will refer to as the Argentona site when talking about the site located at the lower part of the Argentona ephemeral stream.

*"L 85-Please provide the percentile for a 90-mm event here and reference Figure 2a."*

In the new version of the manuscript, we provided the percentile for the 90 mm event in the new as suggested by the reviewer:

**"Three samplings were conducted in the southern section of Maresme County during 2019 and 2020. The two first samplings (hereinafter P1 and P2, chronologically) were performed shortly after an EPE with an accumulated precipitation rate of ~90 mm in one day, which corresponds to the 99.6 wet-day percentile (Fig. 2a)."**

*"L 239: It should be noted this is not necessarily a good assumption, as shown by studies like Weintein et al., 2011; Sawyer et al., 2014, Wong et al., 2020, and many others, but it is understood that it is not very feasible to mobilize a high-resolution sampling effort near the sediment-water interface on the tail of an extreme precipitation event, and this is what would be needed to alleviate the assumption."*

A clearer description of the assumptions and limitations associated with the quantification will be provided throughout the new version of the manuscript (see example below, Section 4.1.3). See also the response to reviewer 1 regarding the endmember selection.

**"Since it was not possible to directly collect the discharging groundwater, by using onshore samples we are implicitly assuming that no nutrient transformation occurred between the sampling point and the discharge point, within the subterranean estuary (Cook et al., 2018). This assumption is perhaps one of the main sources of uncertainty in the reported nutrient fluxes as it has already shown by many authors (Sawyer et al., 2014; Weinstein et al., 2011; Wong et al., 2020). It should also be noted that these SGD-derived nutrient estimates may be biased due to the groundwater endmember selection, since nutrient concentrations in discharging groundwaters may vary during EPE due to dilution, increasing lixiviation of fertilizers, or enhancement of biogeochemical reactions in the mixing zone of coastal aquifers** (Spiteri et al., 2008)**. Although all the assumptions made for nutrient fluxes quantification may result in high degrees of uncertainty, the results presented in this study enable the assessment of EPE significance as a major driving force transporting nutrients to the coastal ocean."**

"*Figure 5: Rather than showing the portion of nutrient fluxes attributed to terrestrial and marine SGD with bullseyes, consider coloring the bars below directly (i.e. stacked dark and light blue bars) to condense the information into one graphic style.*"

Figure 5 has been modified in order to separate the pie charts and the bar plot in two subplots. However, we decided not to use stacked bar plots because the logarithmic scale hamper the recognition of the percentages.